# Yolk-deprived *Caenorhabditis elegans* secure brood size at the expense of competitive fitness

Ellen Geens[1], Pieter Van de Walle[1], Francesca Caroti[2], Rob Jelier[2], Christian Steuwe[2], Liliane Schoofs[1], Liesbet Temmerman[1]

**Oviparous animals support reproduction via the incorporation of yolk as a nutrient source into the eggs. In *Caenorhabditis elegans*, however, yolk proteins seem dispensable for fecundity, despite constituting the vast majority of the embryonic protein pool and acting as carriers for nutrient-rich lipids. Here, we used yolk protein–deprived *C. elegans* mutants to gain insight into the traits that may yet be influenced by yolk rationing. We show that massive yolk provisioning confers a temporal advantage during embryogenesis, while also increasing early juvenile body size and promoting competitive fitness. Opposite to species that reduce egg production under yolk deprivation, our results indicate that *C. elegans* relies on yolk as a fail-safe to secure offspring survival, rather than to maintain offspring numbers.**

## Introduction

Through evolution, multiple modes of reproduction have been established in the animal kingdom, ranging from hydrozoan asexual reproduction to mammalian viviparity. In this spectrum, many species rely on oviparity. Central to the production of yolk in numerous egg-laying species is vitellogenesis (Sappington & Raikhel, 1998; Brawand et al, 2008; Carducci et al, 2019). Here, the proteins are produced that act as carriers for sugars and lipids to make up the glyco-lipoprotein mixture of egg yolk (Wallace, 1985). Expression of the yolk protein precursors or vitellogenins takes place in somatic tissues, for example, the vertebrate liver (Wallace, 1985), insect fat body (Tufail et al, 2014), or nematode intestine (Kimble & Sharrock, 1983), after which they are secreted into circulation. When confronted with a decrease in vitellogenins, many oviparous organisms, including invertebrates like *Drosophila melanogaster* (Bownes et al, 1991) and *Cimex lectularius* (Moriyama et al, 2016), reduce offspring numbers, highlighting the importance of yolk proteins for reproduction. However, observations in *Caenorhabditis elegans* show that not all egg-laying species rely on yolk proteins to maintain offspring production (Grant & Hirsh, 1999; Van Rompay et al, 2015; Ezcurra et al, 2018; Dowen, 2019).

The *C. elegans* genome contains six vitellogenin genes (*vit-1* to *-6*) which encode four yolk proteins (Fig S1, Sharrock et al, 1990). These proteins are produced in the intestine and secreted into the pseudocoelom, from where the oocytes take them up via receptor-mediated endocytosis. Only one yolk receptor has been identified in *C. elegans*, namely, RME-2 (receptor-mediated endocytosis), a member of the low-density lipoprotein receptor superfamily. This transmembrane protein is expressed in the worm's oocytes, most prominently so in the three most proximal ones, coinciding with the place of yolk protein uptake (Grant & Hirsh, 1999). To ensure that vitellogenesis occurs in the correct place at the correct time, multiple environmental and physiological factors influence *vit* expression (Perez & Lehner, 2019). In earlier work, we identified the transcription factor CEH-60 (*C. elegans* Homeobox) as a regulator of vitellogenesis in *C. elegans*. Despite the near-absent yolk protein pool in loss-of-function mutants of *ceh-60*, overall offspring numbers produced by these worms remain unaltered (Van Rompay et al, 2015). This is in line with observations of others that fecundity of *C. elegans* indeed seems unaffected or limited affected by yolk protein deprivation (Grant & Hirsh, 1999; Ezcurra et al, 2018; Dowen, 2019). Considering that YP170 alone is already responsible for a quarter of all de novo protein synthesis in the intestine of adult worms (Kimble & Sharrock, 1983), the question can be raised as to what extent investing energy and resources in the massive production of yolk proteins and their transport into the oocytes is beneficial to *C. elegans*, if not to sustain production of viable offspring?

Here, we tackle this question by a molecular and physiological comparison of yolk protein–deprived offspring versus yolk protein–provided offspring. Our work shows that yolk protein–deprived mutant worms produce offspring whose embryonic development proceeds more slowly and of which the body size is smaller at early life. Although these phenotypic differences in yolk protein–deprived offspring appear small, we did show that massive yolk protein provisioning procures an advantage to the offspring. When introducing a food shortage, the ability of yolk protein–deprived worms to survive and compete with yolk protein–provided

[1]Department of Biology, KU Leuven, Leuven, Belgium  [2]Department of Microbial and Molecular Systems, KU Leuven, Leuven, Belgium

Correspondence: liesbet.temmerman@kuleuven.be

animals decreases. Overall, our results show that yolk proteins are not necessarily used directly by all oviparous animals to maintain their fecundity, but in some species, like *C. elegans*, rather indirectly to provide offspring with a competitive advantage.

## Results

### Yolk deprivation slows down embryonic development

Embryonic development is characterized by a massive increase in cell number and diversity, resulting from numerous consecutive cell divisions. We hypothesized that early embryogenesis might be directly affected by the assumed decrease in energy availability due to maternal yolk protein deprivation. To allow tracking of cell nuclei during embryogenesis, yolk protein–deprived *ceh-60* mutants were crossed with animals expressing histone–GFP fusion proteins (Murray et al, 2006). Embryonic cell divisions were followed starting from the division of the ABa/p cells until the D cell division. At this time point, all founder cell lineages (AB, MS, E, C, D, and P4) have been established (Sulston et al, 1983). Although the delay in development is more pronounced for *ceh-60(lst491)* compared with *ceh-60(lst466)* embryos, we observed that all cells needed more time to complete their cell cycle during the early stages of embryogenesis (Figs 1A and S2). These results suggest that a lack of functional CEH-60 provokes—be it directly or indirectly—a general delay in embryonic development that may be sustained beyond the early embryonic cell divisions observed here. Indeed, in follow-up light microscopic observations of embryonic development until hatching, the time needed for *ceh-60* embryos to complete their development increased by 47 min *(lst466)* and 22 min *(lst491)* in comparison to WTs (Fig 1B). As an additional control, we quantified the time needed to reach hatching in *vrp-1(lst539)* mutants which contain yolk protein levels comparable to those of *ceh-60* animals (Van Rompay et al, 2015) and found that they display a similar delay in embryonic development (48 min, Fig 1B). The measured delay in total time until hatching does not seem to result from a specific phase of the embryonic development. Although we observed that the *ceh-60* and *vrp-1* embryos might spend slightly less time going from the comma to 1.5-fold stage (Fig 1B), all other measurements show that these yolk protein–deprived animals overall suffer from a small but consistent delay in embryonic development, indicating that yolk protein provisioning might slightly speed up embryonic development.

### Body size of yolk-deprived animals is only affected in early life

The end of embryonic development in *C. elegans* is defined by hatching of the first juvenile stage (L1). It has been reported that the increase in body size at time of hatching observed in offspring from older parents correlates with an increase in yolk protein provisioning (Perez et al, 2017). Hence, we reasoned that the near abolishment of yolk proteins in *ceh-60* and *vrp-1* mutants may also result in smaller offspring. To determine the body size of these mutants, we measured the length and width of L1 juveniles. Although embryos collected from yolk protein–deprived *ceh-60* and *vrp-1* mutants do not differ in developmental stage from WTs

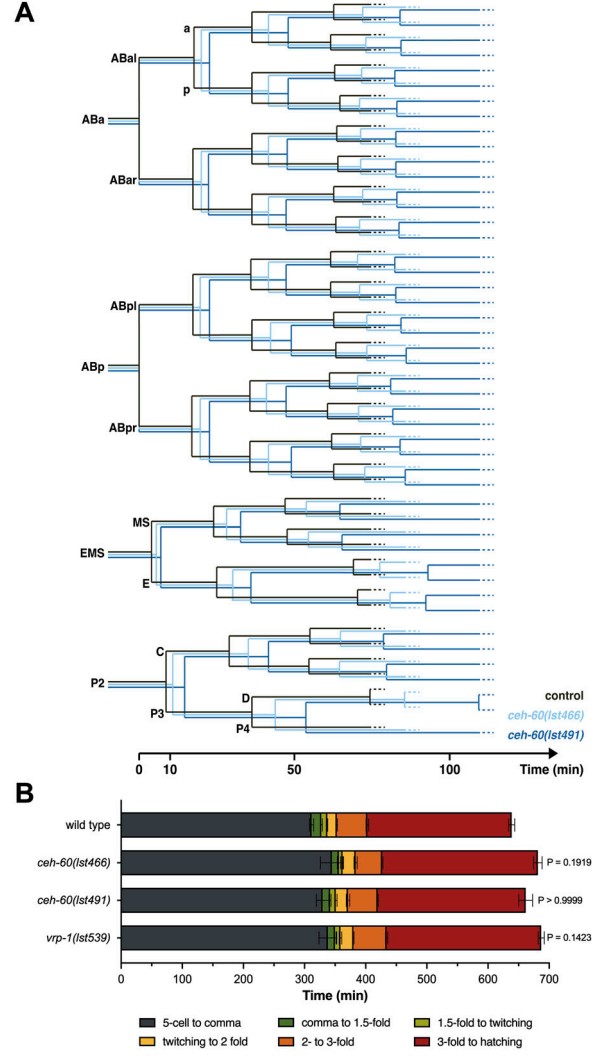

**Figure 1.  Embryonic development is delayed in *ceh-60* and *vrp-1* mutants.**
**(A)** Cell cycle measurements were used to construct an embryonic lineage tree of control, *ceh-60(lst466)* and *ceh-60(lst491)* worms expressing nuclear GFP (data from detailed tracking of three independently imaged embryos per condition). For both *ceh-60* mutants, every tracked cell shows an increase in cell cycle duration in comparison with controls. Timing starts at the division of the ABa/p cells and observation continued until the D-cell division. After a founder cell has been born, all corresponding daughter cells are named based on their position across the anterior–posterior axis. The anterior (a) daughter cell is placed at the top of the branch, whereas the posterior (p) is placed at the bottom. Detailed statistical analysis can be found in Fig S2. **(B)** Overall, the embryonic development of the yolk protein–deprived *ceh-60(lst466)*, *ceh-60(lst491)*, and *vrp-1(lst539)* mutants exhibit a small, although not significant, delay in comparison to WT development. Indication of recognizable embryonic stages reveals that the yolk protein–deprived worms consistently require more time to reach a next stage, except when passing from the comma to 1.5-fold stage. *P*-values at the end of the bar plot compare the time needed for the embryo to hatch starting from the 5-cell stage (two-sided Kruskal–Wallis test with Dunn's post hoc test, N ≥ 5). Error bars represent standard error of mean. Source data are available for this figure.

(Fig S3), the L1 offspring were significantly smaller in both length and width compared with WT L1s (Fig 2A). Consequently, yolk protein provisioning in *C. elegans* does not only affect the body size at hatching (Perez et al, 2017) but also further into the first juvenile

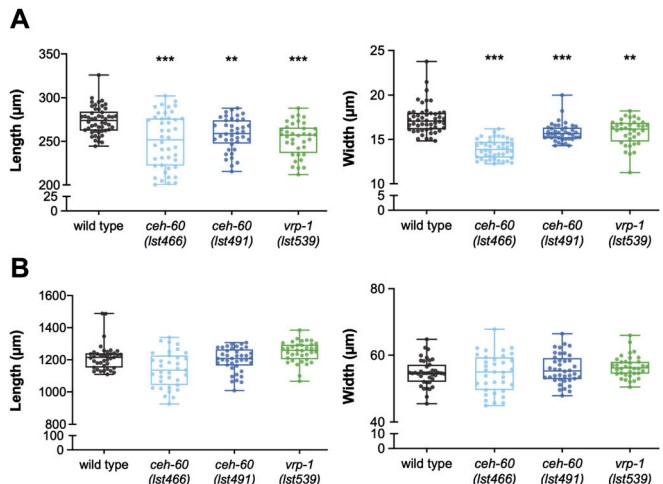

**Figure 2.   The body size defect found in L1 juveniles of yolk protein–deprived *ceh-60* and *vrp-1* mutants has disappeared by adulthood.**
**(A)** Both the length (midline) and width (at grinder) of L1 juveniles are significantly decreased in *ceh-60(lst466)*, *ceh-60(lst491)*, and *vrp-1(lst539)*. **(B)** However, *ceh-60* and *vrp-1* mutants have a WT body size when reaching adulthood. Statistical significance compared with WT was determined using two-sided Kruskal–Wallis test with Dunn's post hoc test (A, B left) and one-way ANOVA with Tukey's post hoc test (B right). **$P ≤ 0.01$, ***$P ≤ 0.001$. N ≥ 35. Source data are available for this figure.

stage. Once the animals have hatched, independent feeding can be started, which frees them from yolk as the only nutrient source. To verify whether the initial decreased body size is maintained into adulthood, we also quantified body sizes of worms on the first day of adulthood. Both *ceh-60* and *vrp-1* mutants overcame the early juvenile size defect by the time they had reached adulthood, indicating that yolk protein deprivation in early life does not lead to a decrease in adult body size (Fig 2B).

### *ceh-60* affects proteins required for uptake and intracellular transport of yolk

Although the impact of mutating *ceh-60* on yolk protein levels in *C. elegans* is massive, the observed delay in embryonic development is small at best (Fig 1). Moreover, contrary to findings in other organisms, *C. elegans*' brood size does not correlate with the level of yolk protein availability (Van Rompay et al, 2015; Ezcurra et al, 2018). These observations hint at compensatory mechanisms to overcome yolk protein deprivation. We reasoned these could involve backup by other imported proteins and/or an oocytic response to try and improve yolk import. To shed light on these options, we performed a differential proteomics experiment comparing embryos of WTs with those of two *ceh-60* mutants. Both the *ceh-60(lst466)* (early stop) and *ceh-60(lst491)* (splice site acceptor) alleles are known to result in yolk protein deprivation (Van Rompay et al, 2015; Van De Walle et al, 2019), whereas leading to a distinct outcome for at least one phenotype, that is, cuticle permeability (Fig S4). We reasoned that differential results that are shared between strains carrying these alleles would help focus our work toward changes that are relevant to understanding yolk protein deprivation. We identified 837 proteins,

of which 198 were significantly increased and 34 decreased in *ceh-60(lst466)*, whereas in *ceh-60(lst491)*, 41 proteins were significantly increased and 189 decreased versus WT controls (fold change ≤ 0.80 or ≥ 1.20 and $P < 0.05$). Overall, the *lst466* allele clearly induces more protein levels to be increased in comparison to WT, whereas the inverse is true for the *lst491* allele (Fig S5A). Although this is an interesting finding in and of itself in light of the described inhibition-to-activation switch of the CEH-60 transcription factor complex (Dowen, 2019), we here focused on proteins that were increased or decreased in both mutants to probe for responses to yolk protein deprivation in *ceh-60* embryos.

In total, 25 proteins were detected at lower levels in mutant embryos of both alleles (Table S1). Gene Ontology (GO) analysis of these proteins identified statistically overrepresented biological processes linked to lipid transport (Fig S5B). This matched expectations since all six VITs, responsible for lipid uptake into embryos, are strongly reduced in *ceh-60* embryos (Table S1). In line with previous observations for another *ceh-60* allele (Dowen, 2019), we observed a significant decrease in lipid content in the embryos of *ceh-60(lst466)* and *ceh-60(lst491)* mutants. According to expectations, we also measured a similar decrease in another yolk protein–deprived mutant, *vrp-1(lst539)* (Fig S6). Although GO analysis of the 25 less abundantly present proteins only identified processes linked to lipid transport, we could distinguish multiple proteins involved in metabolism that were less abundant in *ceh-60* embryos. These include mitochondrial CoA reductase MECR-1 and mitochondrial carrier protein DIF-1, both involved in fatty acid metabolism, isocitrate dehydrogenase IDHG-1 and succinyl-CoA ligase SUCG-1 which catalyze enzymatic reactions in the citric acid cycle, and GYG-2, an ortholog of human GYG1 involved in glycogen metabolism (Oey et al, 2005; Gurvitz, 2009; Garrett and Grisham, 2010; Stemmerik et al, 2017). In addition, five of the less abundant proteins have already been linked with embryonic morphogenesis. The CAD protein PYR-1 and gap junction component INX-3 are essential for pharyngeal morphogenesis, whereas the Arp2/3 subunit ARX-7, myosin light chain MLC-5, and mitochondrial carrier protein DIF-1 are involved in ventral closure, embryonic elongation, and tissue differentiation, respectively (Ahringer, 1995; Sawa et al, 2003; Starich et al, 2003; Franks et al, 2006; Gally et al, 2009). These proteins are not the focus of our work but provide targets to, in future work, probe for details on genetic mechanisms connecting CEH-60 with the outcome of delayed embryonic development (Fig 1).

Apart from the 25 proteins which were less abundant in both *ceh-60(lst466)* and *ceh-60(lst491)* embryos, we also found 35 proteins to be present at higher levels in these mutant embryos (Table S1). These proteins are prime candidates to deliver insight in mechanisms that may exist in *ceh-60* animals to retain their normal reproductive capacity despite yolk protein deprivation. Hence, we used an unbiased approach, wherein we quantified the offspring production of worms submitted to specific knockdown of these targets. Although down-regulation of a few genes (*erh-2*, *vps-60*, *rpl-1*, *dohh-1*, *snx-3*, and *cope-1*, Fig S5C) did lead to a consistent decrease in early reproduction of *ceh-60* mutants in comparison with WTs, no process was revealed in which the genes shared a role that would provide new mechanistic insights. GO analysis of proteins that are more abundantly present in *ceh-60* mutants,

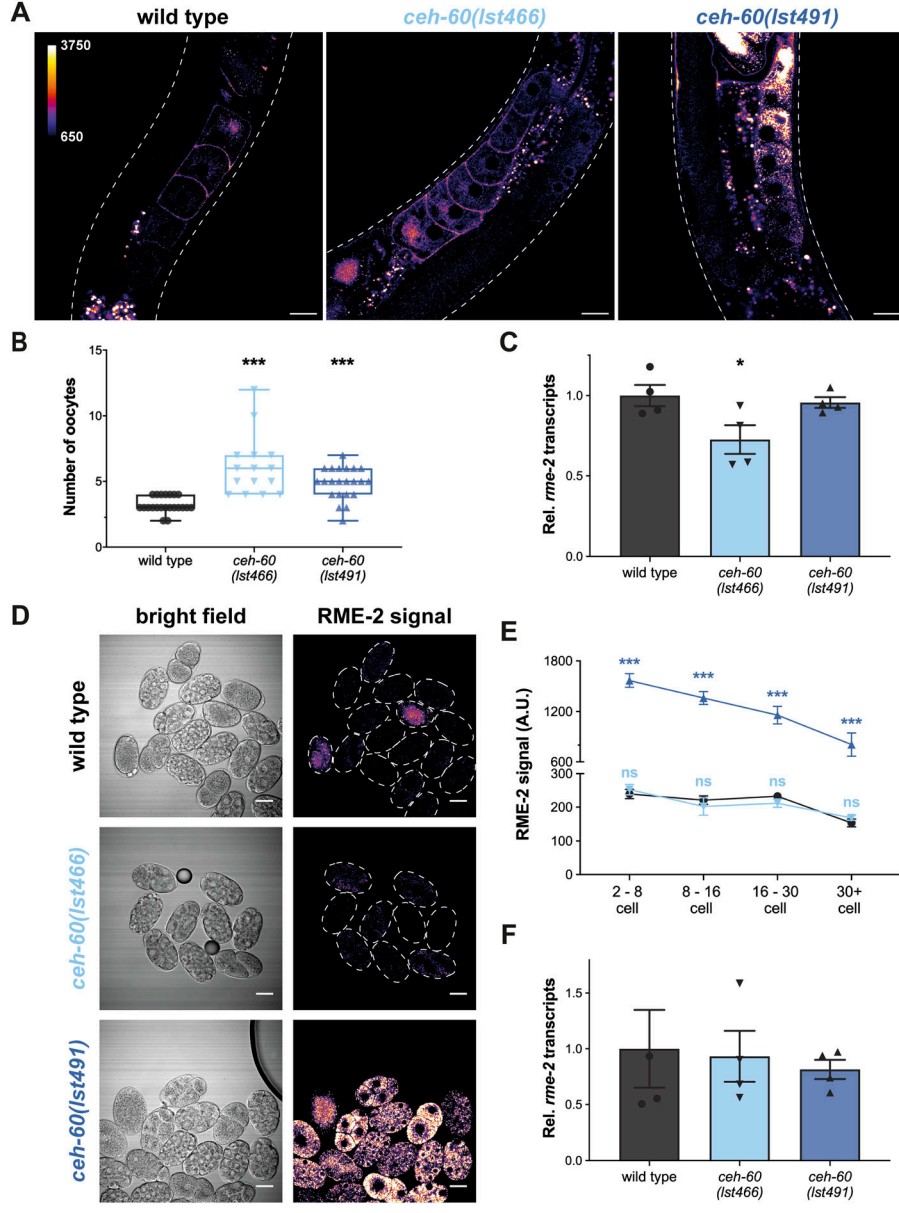

**Figure 3. *ceh-60* affects RME-2 levels and localization in oocytes and embryos.**
**(A)** Fluorescent images of adult worms expressing *rme-2p::rme-2::GFP::rme-2 3'UTR* show that in WT and *ceh-60(lst466)* oocytes, RME-2 is primarily located at the plasma membrane, whereas a *ceh-60(lst491)* mutation leads to RME-2 signal in the cytoplasm. **(A, B)** However, oocytes of both *ceh-60* mutants express the yolk protein receptor RME-2 in younger oocytes compared with WT worms. **(C)** Using quantitative RT–PCR, quantification at the RNA level showed that *ceh-60* mutants do not contain more *rme-2* transcripts, that is, they do not increase expression of the yolk protein receptor. **(D)** Fluorescence of WT, *ceh-60(lst466)* and *ceh-60(lst491)* embryos and **(E)** signal quantification revealed that RME-2 fluorescence in *ceh-60(lst491)* (▲) embryos is maintained at high levels as embryogenesis continues, compared with near-absence of this signal in WT (●) and *ceh-60(lst466)* (▼) embryos. **(F)** However, *rme-2* transcript levels (measured via qRT-PCR) in *ceh-60(lst491)* embryos do not differ from those of WT and/or *ceh-60(lst466)* embryos. Fluorescent images are pseudocolored by pixel intensity, with calibration bar in (A) valid for all fluorescent images. Scale bar 20 μm. White dashed lines show the imaged worm's or embryo's outline. Statistical significance compared with WT in (B) was determined using two-sided Kruskal–Wallis with Dunn's post hoc test (N ≥ 15). In (C, F), statistical significance compared with WT was determined using one-way ANOVA and Tukey's post hoc test, based on four independent experiments with ≥ 1,450 and 13,500 individuals for (C) and (F), respectively. In (E), statistical significance compared with WT was determined using two-way ANOVA with Benjamini–Hochberg post hoc test (N ≥ 7) ns = not significant. *P ≤ 0.05; ***P ≤ 0.001. A.U., arbitrary unit; Rel., relative.
Source data are available for this figure.

however, show that "intracellular transport" is an overrepresented biological process in both mutants (Fig S5B). To study transport of yolk proteins specifically, we studied the yolk receptor RME-2 in oocytes and embryos using a GFP-tagged fusion protein (Balklava et al, 2007).

In agreement with Grant and Hirsh (1999), all (N = 21) control animals show RME-2 signal near the plasma membrane of oocytes. Although RME-2 in control worms is predominantly present in the three most proximal oocytes, both *ceh-60* mutants express the yolk protein receptor also in more distal ones (Figs 3A and B). Despite both *ceh-60* mutants expressing RME-2 in younger oocytes, we could also observe major differences in RME-2 localization between worms carrying the *lst466* or *lst491* mutant allele. Comparable to control animals, RME-2 in *ceh-60(lst466)* animals (N = 15) is

concentrated at the plasma membrane, whereas in *ceh-60(lst491)* oocytes (N = 14) RME-2 is scattered around the cytoplasm (Fig 3A). To probe for RNA-level differences in these strains, we quantified *rme-2* transcripts in oocytes and embryos that are contained within adults and found that these are significantly reduced in *ceh-60(lst466)* but not in *ceh-60(lst491)* (Fig 3C).

After fertilization, RME-2 accumulates in vesicles in the embryo's cytoplasm and quickly disappears as the first embryonic cell divisions take place (Grant & Hirsh, 1999). We collected embryos isolated from the hermaphrodites' gonads and compared RME-2 reporter signal for different embryonic stages. The RME-2 fluorescent signal is already low in young control and *ceh-60(lst466)* embryos, whereas RME-2 is still abundantly present in young and older *ceh-60(lst491)* embryos alike (Figs 3D and E). This is likely due

to a (post)translational cause because *rme-2* transcript levels in these embryos are not different from those of WT or *ceh-60(lst466)* (Fig 3F).

## Abundant YP170 increases postembryonic starvation survival

Evidence to date implies that severely diminished yolk protein production, including in *ceh-60* mutants, does not impact the fecundity of *C. elegans* (Van Rompay et al, 2015; Ezcurra et al, 2018; Dowen, 2019). However, a reduction in maternal vitellogenin provisioning delays L1 starvation recovery due to a delayed blast cell division that would normally take place in the L1 stage (Olmedo et al, 2020). L1 juveniles provided with high levels of yolk proteins are not only able to recover faster from starvation but also are better protected against postembryonic starvation in general (Chotard et al, 2010; Van Rompay et al, 2015). Remarkably, the level at which *ceh-60(lst466)* and *vrp-1(lst539)* juveniles are impacted by starvation differs widely despite their similar yolk protein pool (Van Rompay et al, 2015). To probe for the presence of an additional effect of *ceh-60* mutations on juvenile starvation survival, we compared the survival of *ceh-60(lst491)* juveniles with that of WT, *ceh-60(lst466)*, and *vrp-1(lst539)* juveniles under starvation (Fig 4A). In comparison to WT juveniles with a mean survival of ~12 d, the yolk protein–deprived *ceh-60(lst466)*, *ceh-60(lst491)*, and *vrp-1(lst539)* L1s all have a significantly reduced survival of ~3, 7, and 8 d, respectively (all *P* < 0.001). Confirming our previous research (Van Rompay et al, 2015), starvation survival of *ceh-60(lst466)* juveniles is significantly lower than that of *vrp-1(lst539)* juveniles (*P* < 0.001; Fig 4A). However, no difference could be found between *vrp-1(lst539)* and *ceh-60(lst491)* juveniles. We therefore propose that the yolk protein deprivation in *ceh-60* and *vrp-1* mutants leads to a decline in L1 starvation survival similar to the one measured in *ceh-60(lst491)* and *vrp-1(lst539)* juveniles, and that the almost abolished survival of *ceh-60(lst466)* is probably caused by an additional defect in these animals.

While they both suffer a significant lack of yolk proteins, other phenotypes of the two *ceh-60* mutants used here differ significantly, including some directly related to reproduction:

postembryonic starvation survival (Fig 4), embryonic development (Fig 1), and *rme-2* expression (Fig 3). Hence, we created new mutants to directly probe for the role of specific yolk proteins on juveniles' capacity to withstand starvation. Given that six *vit* genes are expressed in the *C. elegans* intestine forming four different yolk proteins (YP170A, YP170B, YP115, and YP88, Fig S1) and knowing that compensatory mechanisms exist between different yolk proteins in *C. elegans* (Sornda et al, 2019), it is unclear whether all these yolk proteins are necessary to survive L1 starvation. To unveil the contribution of individual yolk proteins to L1 starvation survival, we created mutants in which progressively more *vit* genes are deleted using CRISPR/Cas9. Because of the high sequence similarity of *vit-3, -4*, and *-5* (99%) and the presence of multiple open reading frames between *vit-3* and *vit-4*, no CRISPR RNAs (crRNAs) could be designed to specifically knockout *vit-3, -4*, or *-5* without affecting other (*vit* or other) genes at that locus. Therefore, in addition to genetic removal of YP170B established by *vit-2 vit-1* knockout, further removal of YP170A (encoded by *vit-3, -4*, and *-5*) was accomplished by *vit-5* RNAi which causes knockdown of all *vit* genes except *vit-6* (Ezcurra et al, 2018). Abolishment of the YP170B pool alone leads to a significant decrease in mean L1 starvation survival from ~12 d in WTs to ~10 d (*P* < 0.001) in mutants. Additional removal of YP170A causes an even greater decrease in mean survival to ~7 d (Fig 4B). Assuming that parental yolk protein content is indicative of the yolk protein content present in the offspring, YP170B-deprived animals (*vit-2(lst1671) vit-1(lst1678)*) still contain more YP170 than YP170A&B-deprived worms (*vit-5* RNAi–treated *vit-2(lst1671) vit-1(lst1678)*) (Fig S7). These results suggest that the availability of YP170 might determine a juvenile's capacity to survive starvation. In contrast, additional elimination of YP115 and YP88 caused by *vit-6* deletion did not affect larval survival any further (*P* = 0.607), suggesting that these two yolk proteins do not play a central role in L1 starvation survival (Fig 4B). Alternatively, these results may be considered at the level of the yolk protein complexes, with YP170A, YP115, and YP88 constituting complex A, and complex B being a YP170B dimer (Sharrock et al, 1990, Fig S1). In absence of their interaction partner, YP170A, YP115 and YP88 may not be able to form a yolk protein complex. Thus, even if a functional *vit-6* gene is present, the entire A

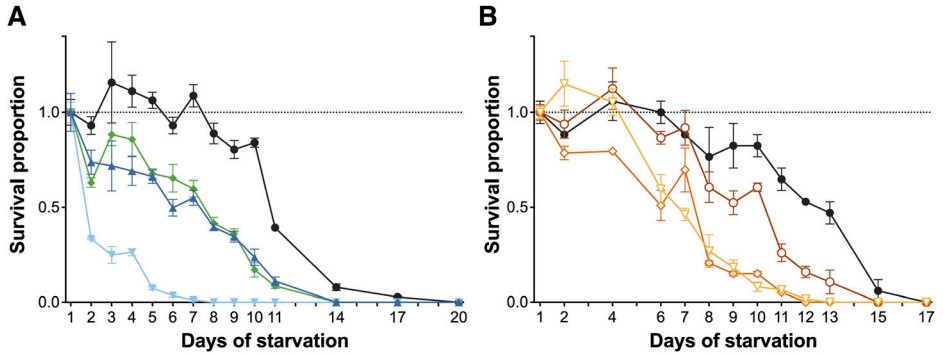

**Figure 4. Survival of postembryonic starvation is affected in yolk protein–deprived L1 juveniles.**
**(A)** Compared with WTs (●), the survival curves of *ceh-60(lst466)* (▼), *ceh-60(lst491)* (▲), and *vrp-1(lst539)* (◆) L1 juveniles were significantly different (*P* < 0.001) when kept in complete absence of food. **(B)** Under the same conditions, decreasing YP170 levels lead to defective survival as apparent by the significant differences found in survival curves between WT (●) and YP170B-deprived (○) (*vit-2(lst1671) vit-1(lst1678)*; *P* < 0.001), YP170(A&B)-deprived (◇) (*vit-5* RNAi–treated *vit-2(lst1671) vit-1(lst1678)*; *P* < 0.001), and total YP-deprived L1s (▽) (*vit-5* RNAi–treated *vit-6(lst1667); vit-2(lst1671) vit-1(lst1678)*; *P* < 0.001).

Survival curves of YP170-deprived and total yolk protein–deprived L1s were not significantly different from each other, suggesting that *vit-6* deletion does not determine the capacity of L1 juveniles to survive postembryonic starvation (*P* = 0.607). Statistical analysis was performed using a log-rank test of smoothed survival curves, based on averages of three independent replicates. Error bars represent standard error of mean.
Source data are available for this figure.

complex of yolk proteins may fail to contribute to the survival of L1 starvation in YP170A-deprived animals. Together, our results show that removal of one yolk protein, YP170B, already significantly impacts the ability of *C. elegans* to survive juvenile starvation, whereas abolishment of the YP170 protein pool (Fig 4B) leads to a mean survival comparable to that of mutants of vitellogenesis regulators (Fig 4A).

### Severe yolk protein deprivation leads to a competitive disadvantage

Although yolk protein–provided and yolk protein–deprived worms produce similar numbers of offspring (Van Rompay et al, 2015; Ezcurra et al, 2018; Dowen, 2019), our results show that several yolk protein–deprived mutants produce offspring with reduced fitness as indicated by their slower embryonic developmental speed (Fig 1), smaller juvenile body size (Fig 2), and decreased ability to survive starvation upon hatching (Fig 4). Although these phenotypic differences do not lead to an observably reduced fecundity (Van Rompay et al, 2015), we hypothesized that competition between yolk protein–deprived and yolk protein–provided animals may unveil a competitive advantage for yolk protein–provided worms. Based on the study of Crombie et al (2018), we initiated competition by pairing a non-fluorescent focal worm with a fluorescently labelled competitor (*rps-0p::roGFP2-Orp1*) on a transient bacterial food patch for 7 d, which corresponds to approximately two generations. The competitive index (CI, see the Materials and Methods section) was used as a measure of fitness of the focal strain in comparison with the competitor.

Fluorescent *rps-0p::roGFP2-Orp1* animals are suitable competitors because the $\log_2(\text{CI})$ value when competing with WT is not significantly different from 0 ($P = 0.5837$ [one sample *t* test], Fig 5A), indicating that no competitive difference between the competitor and WT strain could be found. Mutation of *ceh-60* (*lst466* or *lst491*) or *vrp-1* reduces all yolk proteins (Van Rompay et al, 2015; Dowen, 2019; Van De Walle et al, 2019 and this work) and leads to a clear competitive disadvantage (Fig 5A). It therefore seems that worms with a severely impaired yolk protein pool suffer competition defects in comparison with yolk protein–provisioned animals. Given this result, it can be asked whether the competitive fitness of an animal depends on different contributions of the distinct yolk proteins, as seemed to be the case in L1 starvation assays (Fig 4). Because in this experimental setup the focal and competitor strains are kept on the same food patch, the competitor strain needs to be made insensitive to *vit-5* RNAi by feeding, which is used to clear YP170 from the focal strain. For this, we crossed the competitor strain with *sid-1(pk3321)* mutant worms, which causes resistance to RNAi by feeding (Winston et al, 2002; Tijsterman et al, 2004). This did not influence the competitive fitness of the GFP-marked competitor worms ($P = 0.1387$ [one sample *t* test], Fig 5B). Elimination of only YP170B (genetically via *vit-2 vit-1* mutation) did not affect the worms' competitive fitness. However, additional removal of YP170A via *vit-5* RNAi–mediated knockdown did lead to a competitive disadvantage in comparison to WT (Fig 5B). This decrease in competitive fitness was not aggravated by elimination of YP115 and YP88 via additional *vit-6* knockout (Fig 5B). Together with the

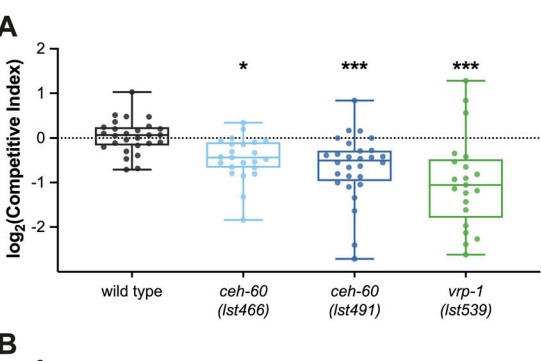

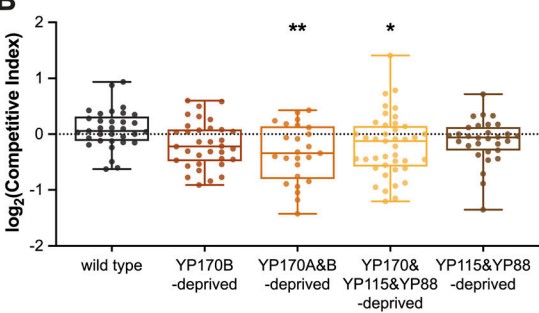

**Figure 5. Total YP170 matters to competitive fitness of *C. elegans*.**
**(A)** Mutants of vitellogenesis regulators *ceh-60* and *vrp-1* exhibit a competitive disadvantage in comparison to yolk protein–provisioned WTs. **(B)** Abolishment of all YP170 (*vit-5* RNAi–treated *vit-2(lst1671) vit-1(lst1678)*) and of all yolk proteins (*vit-5* RNAi–treated *vit-6(lst1667); vit-2(lst1671) vit-1(lst1678)*), leads to a significant decrease in competitive fitness compared with WTs. In contrast, removal of YP115 and YP88 in *vit-6(lst1667)* animals, or of YP170B alone in *vit-2(lst1671) vit-1(lst1678)* worms, does not affect the competitive fitness of *C. elegans*. Statistical significance compared with WT was determined using two-sided Kruskal–Wallis with Dunn's post hoc test. *$P \leq 0.05$, **$P \leq 0.01$, ***$P \leq 0.001$. N ≥ 21.
Source data are available for this figure.

fact that deletion of *vit-6* alone does not lead to a competitive disadvantage (Fig 5B), we can conclude that YP115 and YP88 are dispensable for competitive fitness. Because the bacterial diet of *C. elegans* can influence multiple life history traits, including metabolism, we confirmed that the absence of a decreased competitive fitness in YP170B- and YP115/YP88-deprived worms does not depend on whether HT115 or OP50 *E. coli* are used in the assay (Fig S8). Overall, in contrast to L1 starvation survival assays where the removal of merely one yolk protein (i.e., YP170B) already has a significant impact on survival, only abolishment of the whole YP170 pool lowered competitive fitness in our assays.

## Discussion

Yolk is generally deemed essential to the reproduction of all oviparous species. In contrast to several egg-laying animals that lower their fecundity when yolk proteins are limited, multiple studies in *C. elegans* reported that this nematode could maintain its reproductive success despite yolk protein deprivation (Grant & Hirsh, 1999; Van Rompay et al, 2015; Ezcurra et al, 2018; Dowen, 2019). Given their major role in the makeup of yolk (this work and the works by Perez et al, 2017; Dowen, 2019; Van De Walle et al, 2019;

Kern et al, 2021), these observations nuance the central role of yolk for fecundity in *C. elegans*. However, which benefits *C. elegans* might obtain by investing in vitellogenesis or massive yolk protein production, remained largely unclear. Here, we show that although yolk protein–deprived offspring are viable, their fitness is lower than that of yolk protein–provisioned animals. We observed that yolk protein deprivation may already impact the development of *C. elegans* as early as the first embryonic cell divisions, although the effect is small. Because the fecundity of yolk-deprived animals is unaltered, the impact of severe yolk protein deprivation in *C. elegans* can be considered limited under standard laboratory conditions (this work and the works by Chotard et al, 2010; Van Rompay et al, 2015; Perez et al, 2017; Dowen, 2019; Van De Walle et al, 2019). However, we showed that when *C. elegans* is challenged by absence of food at hatching or by introducing a yolk protein–provisioned competitor, so is the survival of yolk protein–deprived worms.

It is known that the postembryonic developmental rate in *C. elegans* correlates with yolk protein provisioning (Perez et al, 2017), suggesting that higher levels of yolk proteins do confer some advantage to the offspring. In line with these findings, we observed that yolk protein–deprived *ceh-60* and *vrp-1* mutant embryos exhibit a small but consistent delay in development that already affects the time of hatching (Fig 1). This observation is not only valid for this invertebrate; embryonic development in zebrafish correlates with vitellogenin levels, and the same is true for yolk levels in natricine snake eggs (Aubret et al, 2017; Yilmaz et al, 2018). In the "boom-and-bust" lifestyle of *C. elegans* (Frézal & Félix, 2015), fast development is likely under strong selection pressure. For a worm that produces offspring that develop faster, food availability at hatching may still meet the offspring's needs. If no food is available upon hatching, fast embryonic development could allow well-provisioned juveniles to leave the unfavorable environment before slower developing offspring. Assuming no trade-offs for locomotion, this should give them a head start in the quest for food. During embryogenesis, an enormous rise in cell number takes place in a short period of time. At 20°C, a *C. elegans* zygote can develop into a juvenile consisting of 671 cells in 800 min (~13.3 h) (Sulston et al, 1983). In lecithotrophic embryos, oxidation of fatty acids originating from yolk lipids leads to the production of acetyl coenzyme A (acetyl-CoA), used in the tricarboxylic acid cycle to produce ATP (Garrett and Grisham, 2010). An impaired ATP production in zebrafish, due to lipolysis inhibition, leads to a lower rate of embryonic development (Dutta & Sinha, 2017). Given that *ceh-60* embryos contain ~50–60% fewer lipids than WT embryos (Fig S6), we hypothesize that ATP availability in mutant embryos might be decreased. In addition, the enzymes IDHG-1 and SUCG-1, both involved in ATP production (Garrett and Grisham, 2010), are significantly down-regulated in *ceh-60* mutant embryos (Table S1). As opposed to lipids, glucose uptake in *C. elegans* oocytes does not rely on the yolk receptor RME-2 but can be imported via the transporter FGT-1 (facilitated glucose transporter) (Kitaoka et al, 2016). Such yolk protein–independent ATP source could explain why the decrease in lipids in yolk protein–deprived *ceh-60* embryos (Fig S6) is not proportional to the (smaller) delay in embryonic development (Fig 1). This hints at another possible strategy to take up nutrient-rich molecules into

*ceh-60* embryos and may explain why our proteomics (Table S1) and follow-up experiments (Fig S5) did not point toward concrete candidates for such a mechanism, if it exists.

Although *ceh-60(lst466)* and *ceh-60(lst491)* mutants exhibit differences in the oocytic subcellular localization of RME-2, both mutants express the yolk protein receptor in younger embryos compared with control animals (Fig 3). Moreover, multiple proteins that are more abundantly present in both *ceh-60(lst466)* and *ceh-60(lst491)* embryos compared with WT ones (Table S1) have already been linked to RME-2 functioning (RAB-5 [Grant & Hirsh, 1999], VHA-8 [Choi et al, 2003], DOHH-1, HAD-1, PRP-21, SPCS-3, Q9U241 [Balklava et al, 2007], and RAB-35 [Sato et al, 2008]). Taken together, these observations might indicate that in the absence of yolk proteins, *ceh-60* mutants try to increase the uptake of yolk by initiating RME-2 expression earlier.

By the end of their development, embryos of multiple oviparous species have not consumed all yolk present in the egg. Also, in *C. elegans*, not all yolk is used during embryogenesis, apparent by the evident presence of yolk proteins in the intestine of newly hatched juveniles (Bossinger & Schierenberg, 1996). Consequently, it has been suggested that yolk may play a role in early postembryonic development in this species. We found that L1 juveniles of the yolk protein–deprived *ceh-60* and *vrp-1* mutants are significantly smaller than WT L1s but manage to catch up on body size by the time they reach adulthood (Fig 2). Hence, it seems that yolk protein deprivation affects postembryonic developmental rate as seen by Perez et al (2017), but that body size is only affected in early postembryonic life.

Our work clearly shows that under optimal laboratory conditions, maintenance of *C. elegans* is not affected by yolk protein deprivation in a significant way. However, in nature, an animal is often challenged by its environment. For example, animals have to compete for the same transient food source. We here unveiled that worms characterized by a substantial reduction in yolk proteins are outcompeted by controls when food is in limited supply (Fig 5). Taken into account that yolk protein–deprived juveniles cannot survive prolonged starvation (Fig 4), the observed effects can explain the competitive disadvantage of worms with decreased yolk protein levels. When zooming in on contributions by individual yolk proteins, no correlation was apparent between the size of the defect and the amount of a specific yolk protein present in the worms. Given the yolk protein dose dependency we observed in L1 starvation survival (Fig 4), the absence of such a correlation in our competitive fitness assays was rather unexpected. Whereas this could reflect true biology, it is also possible that practicalities of our assay limited the ability to observe this. For example, in competitive fitness experiments that run several generations or use other setups (e.g., sparsely distributed food patches), titrating yolk proteins could yet unveil a dose dependency.

Whereas it is abundantly clear that *C. elegans* is able to genetically segregate regulation of *vit* levels and offspring numbers, it is much less clear under which naturally occurring conditions it may do so. For example, dietary restriction causes a decrease in *vit* gene expression in *C. elegans* adults, whereas still increasing yolk protein provisioning in the embryos probably due to an associated reduction of embryo production (Seah et al, 2016; Jordan et al, 2019).

It would be very exciting to explore this further, for example, in other experimental setups inspired by natural habitats, making use of titrations of yolk proteins to probe for concentration-dependent plasticity, or based on naturally occurring mutations in vitellogenesis regulatory genes of wild isolates.

Taken together, our results propose a prioritization of offspring number over offspring quality in yolk-deprived *C. elegans*. This distinguishes *C. elegans* from numerous oviparous organisms, including the model organism *D. melanogaster*, that do decrease offspring numbers when facing vitellogenin deprivation (Bownes et al, 1991). When yolk is aplenty in the worms, hermaphrodites are able to produce high numbers of offspring that are well provisioned to deal with a possibly challenging habitat. However, yolk protein deprivation decreases resources available for reproduction, and these can either be concentrated in fewer offspring or be distributed frugally over many. It is likely that the prioritization of offspring numbers over quality is not unique to *C. elegans*, and this notion is supported by observations in at least one vertebrate representative as well: after *vtg1* or *vtg3* knockout, the number of eggs produced per spawn is not affected in zebrafish, but major defects can be found in embryos and larvae of those *vtg* mutant mothers (Yilmaz et al, 2018). By maintaining a high level of fecundity despite yolk protein deprivation, the decrease in fitness of each individual might be a trade-off for sufficient offspring to remain to sustain survival of the species.

# Materials and Methods

## *C. elegans* maintenance

*C. elegans* was cultured at 20°C on nematode growth medium (NGM) seeded with *E. coli* OP50 as food source using standard methods (Lewis & Fleming, 1995), unless stated otherwise. The following strains were acquired from the *Caenorhabditis* Genetics Center (University of Minnesota): N2 Bristol WT, NL3321 *sid-1(pk3321) V*, RT408 *pwIs116 [rme-2p::rme-2::GFP::rme-2 3'UTR + unc-119(+)]* and RW10029 *zuIs178 [his-72p::his-72::SRPVAT::GFP + unc-119(+)]*; *stIs10024 [pie-1::H2B::GFP::pie-1 3'UTR + unc-119(+)]*. JV10 *jrIs10 [rps-0p::roGFP2-Orp1 + unc-119(+)]* was kindly provided by professor B. Braeckman (Ghent University). Other *C. elegans* strains used in this study are: LSC897 *ceh-60(lst466) X*, LSC902 *vrp-1(lst539) IV*, LSC903 *ceh-60(lst491) X*, LSC1866 *ceh-60(lst466) X*; *zuIs178 [his-72p::his-72::SRPVAT::GFP + unc-119(+)]*; *stIs10024 [pie-1::H2B::GFP::pie-1 3'UTR + unc-119(+)]*, LSC1924 *ceh-60(lst491) X*; *zuIs178 [his-72p::his-72::SRPVAT::GFP + unc-119(+)]*; *stIs10024 [pie-1::H2B::GFP::pie-1 3' UTR + unc-119(+)]*, LSC1947 *sid-1(pk3321) V*; *jrIs10 [rps-0p::roGFP2-Orp1 + unc-119(+)]*, LSC1962 *vit-2(lst1671) vit-1(lst1678) X*, LSC1923 *vit-6 (lst1667) IV*; *vit-2(lst1671) vit-1(lst1678) X*, LSC1830 *vit-6 (lst1667) IV*, LSC1973 *ceh-60(lst466) X*; *pwIs116 [rme-2p::rme-2::GFP::rme-2 3'UTR + unc-119(+)]*, and LSC1974 *ceh-60(lst491) X*; *pwIs116 [rme-2p::rme-2::GFP::rme-2 3'UTR + unc-119(+)]*.

## Embryo mounting for imaging

Isolation and mounting of *C. elegans* embryos was performed based on the study by Murray et al (2006). Four to seven young adults (day 1 of adulthood) were picked into 50–100 $\mu$l S-basal (5.85 g NaCl, 6 g $KH_2PO_4$, 1 g $K_2HPO_4$ in 1 liter milliQ) or M9 buffer (3.0 g $KH_2PO_4$, 6.0 g $Na_2HPO_4$, 0.5 g NaCl, 1.0 g $NH_4Cl$ in 1 liter milliQ) in a watch glass. Using two 20-gauge needles (BD Microlance), worms were cut at the vulva, resulting in the release of the embryos. Next, two-cell stage embryos were carefully isolated from the others, and maximally three embryos were pipetted onto the middle of a glass slide (76 × 26 mm). Cold (4°C) 20-$\mu$m polystyrene beads (Polybead Microspheres 20 $\mu$m [Polysciences]) were added ensuring that the total volume of liquid did not exceed 1 $\mu$l. Lastly, a glass coverslip (20 × 20 mm) was lowered onto the sample and the edges were sealed with molten petroleum jelly or paraffin to avoid the sample from drying out during imaging.

## Analyzing embryonic development

*C. elegans'* embryonic development was studied using two different approaches, one to track individual nuclei during early embryogenesis and another to time key events until completion of embryonic development. To visualize early embryonic cell divisions in *ceh-60* worms, LSC897 and LSC903 animals were crossed with RW10029 worms, resulting in *ceh-60* mutants with GFP-tagged histones (LSC1866 and LSC1924). Genotypes were confirmed through sequencing.

Imaging started as soon as possible after mounting, ensuring that the embryos did not yet pass the four-cell stage at the start of data collection. Imaging was performed using an inverted ZEISS LSM 880 equipped with a 24-mW argon laser used for 488 nm excitation of the sample. The microscope's fast AiryScan mode was used for detection. Every minute, a z-stack of each embryo was made with a Plan-Apochromat 63x/1.4 DIC M27 oil immersion objective. These stacks consisted of 25–32 slices with a spacing of 1 $\mu$m. Because the D-cell division (chosen end point of our analysis) occurs ~83 min after the ABa/p divisions in WT embryos (Sulston et al, 1983), each imaging experiment lasted at least 120 min. The temperature of the sample was kept between 19 and 21°C using a temperature-controlled stage. Per strain, imaging of biological replicates always took place on at least two separate days.

Analysis of the stacks was performed as described by Jelier et al (2016). Briefly, time-lapse images were cropped, and noise corruption was reduced by smoothing the images using a Gaussian filter. Afterward, all images were converted from 16-bit to 8-bit. All these steps were performed in Fiji (Schindelin et al, 2012). Resulting images were analyzed using a customized algorithm written in MATLAB (Mathworks) to retrieve information regarding cell cycles during embryogenesis. These results were exported in the StarryNite format (Bao et al, 2006). Errors in the lineage output were manually corrected, and all the cells were named based on the lineage compiled by Sulston and colleagues (Sulston et al, 1983) using custom-made WormDeLux software (Jelier et al, 2016). Based on the measured cell cycle duration of each cell between the ABa/p and D cell divisions, a lineage tree was created using Adobe Illustrator.

In later stages, tracking of embryonic nuclei becomes impractical. However, *C. elegans* embryos are transparent, making it possible to follow their development using bright field microscopy. Complete development of individual embryos was followed by

collecting time-lapse images over a period of 14 h, at 1 image per minute. Here, only the midplane of the embryo was imaged because this suffices to recognize embryonic stages. An inverted ZEISS Axio Observer Z.1 microscope driven by MetaMorph software and equipped with an ORCA-Flash4.0 V2 camera (Hamamatsu) at 40X magnification was used. Analysis started at the onset of the 5-cell stage. For every embryo, the time needed to reach the comma, 1.5-fold, twitching, two fold, three fold, and hatching stages was determined through manual analysis of the image sequences.

### Body size quantification

To determine the body size at the L1 stage, juveniles were obtained from eggs harvested via standard hypochlorite treatment from synchronized day-1 adults (68–72 h after arrested L1 juveniles were placed on food). The eggs were allowed to hatch in the presence of food to avoid L1 arrest. Because embryonic development in *C. elegans* takes ~13.3 h (Sulston et al, 1983), L1 sampling was performed 18 h after hypochlorite treatment to ensure that all embryos completed their development. In contrast to L1s, synchronized adult worm populations were obtained by leaving the eggs to hatch in absence of food leading to L1 arrest. Afterward, the synchronized worm populations were grown until day 1 of adulthood (68–72 h after introduction to food). Experimental animals were mounted on 2% agarose pads and anesthetized using 10 mM tetramisole. Both L1 and adult animals were imaged using a Leica DM6 B microscope running LAS X software at 20X magnification. To analyze the images, the segmented line tool in Fiji (Schindelin et al, 2012) was used. The width of a worm was determined at the pharyngeal grinder, whereas the length was measured along its midline.

### Acridine orange staining

Cuticle permeability was assessed by acridine orange staining as per Xiong et al (2017). Worms were synchronized by picking L4 animals onto NGM plates seeded with *E. coli* OP50. The next day, adults were washed off the plates and stained with 5 μg/ml acridine orange in M9 buffer for 15 min. Afterward, worms were washed three times, mounted on 2% agarose pads, and anesthetized with 10 mM tetramisole. Images were made using a Leica DM6 B microscope, running LAS X software, using a GFP filter at 5X magnification.

### Differential proteomics

To compare the proteome of yolk protein–deprived *ceh-60* mutant and yolk protein–provisioned WT embryos, we relied on differential proteomics using TMTsixplex (Thermo Fisher Scientific) isobaric labelling. Per condition, six samples were taken consisting of embryos harvested from 12 full-grown (~1,500 worms per plate) NGM plates seeded with *E. coli* OP50. Parental worm populations were first synchronized using standard hypochlorite treatment. Once the worms reached day 1 of adulthood (68–72 h after arrested L1 juveniles were placed on food), embryos were harvested using sodium hypochlorite treatment and washed at least five times with S-basal buffer. To ensure embryonic stages did not differ extensively between different samples, harvested embryos were checked

under a microscope. Embryos were pelleted out of the solution via centrifugation for 1 min at 16,000*g*, and buffer was aspirated until 100 μl were left. According to the manufacturer's TMTsixplex protocol, samples were lysed with lysis buffer (10% sodium dodecyl sulfate in 100 mM triethyl ammonium bicarbonate). Protein homogenates were sonicated for 10 s, kept 50 s on ice (five repeats), and centrifuged at 16,000*g* for 10 min at 4°C. Protein concentration of the samples was determined using a bicinchoninic acid assay. For each sample, 100 μg protein extract was digested overnight with 2.5 μg trypsin at 37°C. Peptide digests were dried using an Univapo 150 ECH vacuum concentrator (Uniequip) and stored at –80°C until labelling. Peptides were labelled with TMT label reagents according to the manufacturer's protocol. After an incubation period of 1 h, the reaction was quenched by adding 8 μl of 5% hydroxylamine.

For each mass spectrometry run, three samples of one condition were pooled with three samples of a second condition in equal amounts. In total, three TMT experiments were performed: WT versus *ceh-60(lst466)*, WT versus *ceh-60(lst491)*, and *ceh-60(lst466)* versus *ceh-60(lst491)*. All samples were run on a 2x Q Exactive Plus (Thermo Fisher Scientific) connected to an Eksigent nanoAcquity LC-Ultra system (Waters). 1 μg of total protein of the labelled sample was dissolved in 10 μl 1% acetonitrile in water before it was loaded on an analytic 200-cm μPAC column (PharmaFluidics). Separation was established by a linear gradient from 1% acetonitrile to 40% acetonitrile in water in 120 min with a flow rate of 300 nl/minute. The 2x Q Exactive Plus was set up in a data-dependent MS/MS mode with a scan spectrum range of 350–1,850 m/z and a resolution of 70,000. The dynamic exclusion time was set at 40 s.

Spectra were analyzed in MaxQuant (Cox & Mann, 2008) using six-plex TMT as internal labels with a reporter mass tolerance of 0.003 D. Oxidation and N-terminal acetylation were set as variable modifications, carbamidomethyl as fixed modification, and up to five modifications were allowed per peptide. MaxQuant's default orbitrap settings were used, and identified peptides were mapped to the UniProt *C. elegans* reference proteome (https://www.uniprot.org/proteomes/UP000001940). All other parameters were set to default, except for enabling match between runs. Data analysis and normalization were performed based on the CONSTANd method (Maes et al, 2016) using the full protein pool for normalization. Peptides were retained for analysis when they minimally contained a valid measurement for at least three out of six channels per condition, except if in one condition (e.g., *ceh-60(lst466)*) a peptide was completely absent. In the latter case, it was retained if detected in at least four out of six channels for any other condition (e.g., WT and/or *ceh-60(lst491)*). Protein identifications were accepted when at least two unique peptides could be identified. Statistical comparison of protein abundances (N = 6, corresponding to six TMT labels) between genotypes was performed using two-way ANOVA with Benjamini and Hochberg multiple comparison correction.

### Lipid quantification using dark field microscopy

Dark field microscopy was performed essentially as described by Fouad et al (2017). Synchronized populations were grown until day 1 of adulthood (68–72 h after arrested L1 juveniles were placed on food). Eggs were released from the gonads as described above (cf.

Embryo mounting for imaging), transferred to a 2% agarose pad, and placed under a coverslip sealed to the slide with petroleum jelly. The slide was mounted on an inverted ZEISS Axio Observer Z.1 microscope driven by MetaMorph software. A red LED ($\lambda$ = 640–650 nm) light strip, powered by 12 V power source, was placed around the slide in a rectangular shape. Images were taken using an ORCA-Flash4.0 V2 camera (Hamamatsu) at 20X magnification.

Using the polygon selection tool in Fiji (Schindelin et al, 2012), each embryo was outlined before measuring its mean pixel intensity. To adjust for variations under light conditions between different slides and days, the mean background pixel intensity was determined using the same region of interest outlined by the embryo. The background signal was then subtracted from the mean pixel intensity measured in the corresponding embryo. For representation purposes only, this background subtraction was applied to images from Fig S6. Pixel intensity measurements, however, were performed on raw images.

### RNAi experiments

All knockdown experiments performed in this study relied on standard RNAi by feeding (Timmons & Fire, 1998; Timmons et al, 2001). Worms were grown on NGM plates supplemented with 0.5 mg/ml ampicillin and 1 mM isopropyl-b-D-thiogalactoside (IPTG). These will henceforward be referred to as "NGM RNAi" plates. NGM RNAi plates were seeded with *E. coli* HT115 RNAi bacteria grown overnight at 37°C in LB medium supplemented with 50 µg/ml ampicillin. Unless stated otherwise, RNAi treatment was initiated at the L1 stage and worms were reared on the RNAi bacteria for at least two generations before conducting an assay. All RNAi clones were either obtained from the Vidal (Rual et al, 2004) or Ahringer (Kamath & Ahringer, 2003) RNAi libraries and verified through sequencing before use.

### Screen for defects in early reproduction

Adapting the protocol by Maia et al (2015) to manual screening, the number of viable offspring produced by a day-1 adult hermaphrodite in 24 h was determined. Eggs were harvested from a mixed worm population via hypochlorite treatment and allowed to hatch overnight in S-basal buffer. Synchronized L1 juveniles were pipetted onto NGM plates seeded with *E. coli* OP50. Once the worms reached the L4 stage, they were transferred to NGM RNAi plates seeded with *E. coli* HT115 carrying either the L4440 empty vector plasmid or the same plasmid backbone containing a dsRNA-producing sequence of the gene of interest. After 30 h, three random day-1 adults were picked to individual NGM RNAi plates seeded with bacteria corresponding to the source plates. Worms were left to lay eggs for 24 h after which they were removed from the plates. The number of offspring was counted when they reached the L4 stage.

### RME-2 reporter imaging

To study the *rme-2* expression in a *ceh-60* mutant genetic background, crosses were made between the endogenous RME-2 reporter RT408 (Balklava et al, 2007) and two *ceh-60* mutant strains, LSC897 and LSC903 (LSC1973 and LSC1974). Genotypes were

confirmed through sequencing. To image the worm's oocytes, animals were mounted on 2% agarose pads and anesthetized with 10 mM tetramisole. *rme-2* expression was visualized using an Olympus FluoView 1,000 confocal microscope with a 488 nm laser. All images were taken at 60X magnification and a scan speed of 8.0 µs/pixel. After imaging, the pixel intensity was confined to a range of 650–3,750 arbitrary units to distinguish *rme-2* expression from autofluorescence using Fiji (Schindelin et al, 2012). This operation does not affect the information contained within the images, which is compatible with the purpose of comparing the *rme-2* expression over different images.

Embryos were mounted as described above (cf. Embryo mounting for imaging) and imaged in the same manner as the oocytes. Similar to optical scattering measurements performed on images obtained using dark field microscopy, the mean pixel intensity for each embryo was determined using Fiji (Fouad et al, 2017). For representation purposes only (not for quantification), images in Fig 3 were "Fire" pseudocolored, and the number of pixels that are allowed to be saturated was limited to 0.2% using Fiji plugins.

### Quantitative real-time PCR

Similar to sampling for differential proteomics, per embryonic sample, embryos from nine full-grown NGM plates (~1,500 day-1 adult worms per plate) seeded with *E. coli* OP50 were harvested using hypochlorite treatment and washed at least five times in S-basal buffer. For the germline-focused samples, synchronized L1 larvae were grown until day 1 of adulthood (68–72 h after arrested L1 juveniles were placed on food) after which adults were washed off the plates and sorted using a COPAS Biosorter (Union Biometrica) to avoid the presence of any laid eggs in these samples. All samples were stored at –80°C in S-basal buffer. RNA extraction was performed using an RNeasy Mini Kit (QIAGEN) combined with DNaseI (QIAGEN) treatment according to the manufacturer's protocols. Homogenization of the samples was achieved using a cooled Precellys homogenizer (Bertin Instruments). Next, cDNA was synthesized by reverse transcription (RT) using the PrimeScript RT reagent kit (Takara), and 20 µl RT-PCR reactions were set up using Fast SYBR Green Master Mix (Thermo Fisher Scientific), 10 µM gene-specific primers (Table 1), and 5 µl cDNA. No template control and RT⁻ reactions were included as controls. For each biological sample, all reactions were run in triplicate in a 96-well plate with a QuantStudio 3 Real-Time PCR System (Thermo Fisher Scientific). Based on geNorm analysis, *cdc-42* (cell division cycle), *pmp-3* (peroxisomal membrane protein), *rpb-12* (RNA polymerase II (B) subunit), and Y45F10D.4 were identified as optimal reference genes (Vandesompele et al, 2002). The relative expression level of *rme-2* was subsequently calculated using these four reference genes for normalization.

### CRISPR/Cas9-mediated vit knockout

Based on the study by Paix et al (2017), deletion alleles of *vit-6*, *vit-2*, and *vit-1* were generated using a co-CRISPR strategy with *dpy-10* as a marker. We designed two crRNAs (Table 2) which generate the greatest possible specific deletion in the *vit* gene's open reading

**Table 1. Primer sets (5′ –3′) used for RT-PCR.**

| Gene | Forward primer | Reverse primer |
|---|---|---|
| *cdc-42* Hoogewijs et al (2008) | CTGCTGGACAGGAAGATTACG | CTCGGACATTCTCGAATGAAG |
| *gdp-2* Higashibata et al (2006) | ACCGGAGTCTTCACCACCATC | ACGAACATTGGAGCATCAGCA |
| *pmp-3* Hoogewijs et al (2008) | GTTCCCGTGTTCATCACTCAT | ACACCGTCGAGAAGCTGTAGA |
| *rpb-12* Temmerman et al (2012) | CAGGTCAAGCTCATCTCAAGTCA | TTTTCGGCGTGGCATTCT |
| *tba-1* Hoogewijs et al (2008) | GTACACTCCACTGATCTCTGCTGACAAG | CTCTGTACAAGAGGCAAACAGCCATG |
| Y45F10D.4 Hoogewijs et al (2008) | GTCGCTTCAAATCAGTTCAGC | GTTCTTGTCAAGTGATCCGACA |
| *rme-2* Perez et al (2017) | CATCATCTGGATCGATTCTTATCAG | AGAAGGATTCTGACCTGAGAC |

frames. Young adults were injected as described by Evans (2006) with a mix consisting of 2.5 µl recombinant *C. elegans* codon-optimized Cas-9 protein (15 mg/ml, kindly provided by the Hollopeter laboratory, Cornell University), 2.5 µl tracrRNA (0.17 mol/l, Integrated DNA Technologies [IDT]), 1 µl *dpy-10* crRNA (0.6 nmol/µl, IDT), 0.5 µl *vit* crRNA 1 (0.6 nmol/µl, IDT), 0.5 µl *vit* crRNA 2 (0.6 nmol/µl, IDT), 1 µl *dpy-10* repair template (0.5 mg/ml, Merck), and 1 µl *vit* repair template (1 mg/ml, IDT). After injection, roller or dumpy F1s were transferred to individual NGM plates and allowed to produce F2 progeny. The F1 animals were then screened by PCR for the presence of the desired gene edit. By stepwise deletion of *vit-6*, *vit-2*, and *vit-1*, the *vit-6 (lst1667) IV*; *vit-2(lst1671) vit-1(lst1678) X* mutant was created. To obtain a *vit-2(lst1671) vit-1(lst1678) X* mutant, the *vit-6* deletion was removed from the *vit* triple mutant by outcrossing with WT (N2) worms.

**Postembryonic starvation survival**

L1 starvation survival assays were performed based on Chotard et al (2010). Mixed cultures were synchronized using hypochlorite treatment. Depending on the experiment, worms were reared on NGM agar plates seeded with *E. coli* OP50 or NGM RNAi plates seeded with *E. coli* HT115 carrying either the L4440 empty vector plasmid or the same plasmid backbone containing a *vit-5* target sequence. Once the worms reached day 2 of adulthood (92–96 h after arrested L1 juveniles were placed on food), eggs were collected via hypochlorite treatment and allowed to hatch overnight at 20°C in S-basal buffer. Because survival of L1 starvation depends on population density (Artyukhin et al, 2013), all cultures

were adjusted to 5 worms/µl in a total volume of 3 ml. At every time point, three aliquots of 15 µl were taken from each sample and transferred to individual seeded NGM plates to allow the juveniles to recover from starvation. Approximately 72 h later, the number of surviving animals was counted. L1 juveniles were sampled until all animals had died. Statistical analysis was carried out as described by Lee and Ashrafi (2008). In short, survival curves were smoothed to a non-increasing function of average survival rates. For each condition, the resulting smoothed survival curve was compared with that of the WT using a log-rank test.

**Yolk protein quantification**

Yolk protein quantification was performed using SDS–PAGE with Coomassie staining based on the study by Sornda et al (2019). Worms were synchronized by hypochlorite treatment and grown until day 2 of adulthood (92–96 h after arrested L1 juveniles were placed on food). Once the desired age was reached, 20 animals were transferred into 25 µl S-basal buffer and frozen at −80°C. The next day, 25 µl of 2X Laemmli sample buffer was added and the samples were incubated at 70°C for 15 min in a shaking thermocycler (300*g*). Afterward, the samples were incubated at 95°C for 5 min and centrifuged at 3,300*g* for 15 min. A total of 15 µl of each sample was loaded on a 4–12% Bis–Tris Criterion XT polyacrylamide gel (Bio-Rad) and run for 10 min at 70 V and 70 min at 140 V using XT MOPS as running buffer. Afterward, the gel was stained with Coomassie brilliant blue and destained with a 40% methanol, 10% acetic acid solution. Gel images were collected using a Bio-Rad Gel

**Table 2. crRNAs and repair templates used for CRISPR edits leading to *vit-1(lst1678)*, *vit-2(lst1671)*, and *vit-6(lst1667)*.**

| Gene | crRNA 1 | crRNA 2 | Repair template |
|---|---|---|---|
| *vit-1* | GATTATTATCGCATCTATAGTGG | CGCTTATTAATTCATAAGCTCGG | aggaaattcattgtccattgtccaatcatgaggtcGATCATCATTGCCTGAATTCCGGccggccttttttttcataattttataacttctgct |
| *vit-2* | GATCATCATCGCCTCTCTCGTGG | CCGCCTGCGTCGCTTATTGATTA | tgaaaacagtccaatcacggttcagccatgaggtcGATAATCATCGCGTCACGGGTGGCATAATGATTTagctagaccggcactttatgtaaattgatcattc |
| *vit-6* | TTCATAGCGCTTGCTCTCTTGGG | CCGTCGACCAGAAGTGCGACAAG | tcggtcacttagatcgatcaatcactatgaagttcTTTATCGCCCTCTAGATACCTGAGGTCCGGCAGGttcgactattgaactacctcttcttcacaatcata |
| *dpy-10* | GCUACCAUAGGCACCACGAGGUUUUAGAGCUAUGCU | / | CACTTGAACTTCAATACGGCAAGATGAGAATGACTGGAAACCGTACCGCATGCGGTGCCTATGGTAGCGGAGCTTCACATGGCTTCAGACCAACAGCCTAT Paix et al (2017) |

*dpy-10* was used as a co-CRISPR marker. Homology arms are indicated in lower case

Doc system and analyzed with ImageLab 6.0. Yolk protein bands were identified based on published data and normalized to myosin (Sornda et al, 2019).

### Competitive fitness

Competitive fitness assays were essentially conducted as described by Crombie et al (2018). One L4 animal of the (non-fluorescent) focal strain of interest and the (fluorescent) competitor strain were picked into a well of a 24-well plate filled with 1.5 ml of NGM agar supplemented with 20 $\mu$g/ml nystatin and 50 $\mu$g/ml streptomycin. Depending on the experiment, the NGM agar was seeded with 10 $\mu$l of *E. coli* OP50 or HT115 (carrying either the L4440 empty vector or *vit-5* producing plasmid) grown overnight at 37°C. After a competition period of 168 h at 20°C, the worms were collected by washing each well with 300 $\mu$l M9 buffer and collecting the supernatant into a 96-well plate. Worms were allowed to settle after which they were washed two more times to remove any residual bacteria. Finally, 250 $\mu$l of the buffer was aspirated from the wells, and animals were anesthetized by adding tetramisole to a final concentration of 1 mM. Brightfield and GFP fluorescence images were captured using a Leica DM6 B microscope driven by LAS X software and the number of non-fluorescent focal and fluorescent competitor animals was counted. If a well contained fewer than 10 animals of either strain, that replicate was discarded because the progenitor individual likely died early because of unforeseen circumstances (e.g., injury from picking). The binary logarithm of the competitive index (CI) was calculated for each well using the formula $CI = p/(1-p)$ where $p$ is the proportion of the focal animals. If the $\log_2$(CI) value equals 0, no competitive advantage for either the focal or competitor strain is observed. A positive value indicates that there are more focal than competitor animals, whereas a negative $\log_2$(CI) value indicates a competitive disadvantage for the focal strain.

### Statistical analysis

Statistical analysis was performed using GraphPad Prism software (version 9.0.2). Detailed information on statistical analysis and samples sizes of each experiment has been included in the figure or table legends or in the main text. Briefly, data were first tested for normality via the Shapiro–Wilk test with alpha = 0.05. Afterward, comparison between the effect of a mutation or RNAi treatment to the WT/control condition was performed using one-way ANOVA followed by Tukey's post hoc test. If, however, the data for one of the conditions were not normally distributed, a two-sided Kruskal–Wallis test with Dunn's post hoc was used. When different embryonic cell cycles (Fig S1) or protein quantifications (Table S1 and Fig S5A Source Data) were compared between genotypes, statistical significance was computed using a two-way ANOVA with Benjamini and Hochberg multiple comparison correction. Statistical analysis for the postembryonic starvation survival assays was performed using a log-rank test of smoothed survival curves as described by Lee and Ashrafi (2008). Conditions were considered to be significantly different from each other if the $P$-value < 0.05. Different levels of significance are indicated as ns, not significant; *$P \leq 0.05$; **$P \leq 0.01$; ***$P \leq 0.001$.

## Data Availability

The data underlying Figs 1–5 and S2, S3, S5–S8 are provided in their correspondingly named Source Data file. The complete list of measured proteins in WT, *ceh-60(lst466)*, and *ceh-60(lst491)* embryos can be found in Fig S5A Source Data. Images that support the findings of this study are available from the corresponding author on request. The mass spectrometry proteomics data have been deposited to the ProteomeXchange consortium via the PRIDE (Perez-Riverol et al, 2022) partner repository with the dataset identifier PXD041110.

## Supplementary Information

## Acknowledgements

We are grateful to Prof. G Baggerman and K Schildermans (CfP, UAntwerpen, Belgium) for LC–MS support, A Kieswetter and E Vandewyer (KU Leuven, Belgium) for technical assistance, FJ Naranjo-Galindo and Dr. MB Van Hiel (KU Leuven, Belgium) for assistance with CRISPR design, and Prof. M Roeffaers (KU Leuven, Belgium) for support of lipid measurements (grant G0H0816N). Some strains and reagents were kindly provided by Prof. B Braeckman (UGent, Belgium), Prof. G Hollopeter and E Partlow (Cornell University, USA), and the *Caenorhabditis* Genetics Center (University of Minnesota, USA). This research was supported by FWO grants G095915, G052217N, and G055017N; and KU Leuven grant C16/19/003. P Van de Walle is an SB PhD fellow of the research foundation—Flanders (FWO) (1S00617N).

### Author Contributions

E Geens: conceptualization, data curation, formal analysis, investigation, visualization, methodology, and writing—original draft, review, and editing.
P Van de Walle: conceptualization, formal analysis, investigation, and writing—review and editing.
F Caroti: investigation, methodology, and writing—review and editing.
R Jelier: software, methodology, and writing—review and editing.
C Steuwe: methodology and writing—review and editing.
L Schoofs: conceptualization and writing—review and editing.
L Temmerman: conceptualization, supervision, funding acquisition, methodology, and writing—review and editing.

### Conflict of Interest Statement

The authors declare that they have no conflict of interest.

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
