## [Reviewer comments · Life Science Alliance]

Life Science Alliance

Yolk-deprived *Caenorhabditis elegans* secure brood size at the expense of competitive fitness

Ellen Geens, Pieter Van de Walle, Francesca Caroti, Rob Jelier, Christian Steuwe, Liliane Schoofs, and Liesbet Temmerman
DOI: <https://doi.org/10.26508/lsa.202201675>

Corresponding author(s): *Ellen Geens, KU Leuven*

Review Timeline:	Submission Date:	2022-08-17
	Editorial Decision:	2022-09-28
	Revision Received:	2023-02-27
	Editorial Decision:	2023-03-21
	Revision Received:	2023-03-28
	Accepted:	2023-03-29

Transaction Report:

September 28, 2022

Re: Life Science Alliance manuscript #LSA-2022-01675

Ms. Ellen Geens
KU Leuven
Naamsestraat 59
Leuven 3000
Belgium

Dear Dr. Geens,

Thank you for submitting your manuscript entitled "Yolk-deprived *Caenorhabditis elegans* secure brood size at the expense of competitive fitness" to Life Science Alliance. The manuscript was assessed by expert reviewers, whose comments are appended to this letter. We invite you to submit a revised manuscript addressing the Reviewer comments.

Thank you for this interesting contribution to Life Science Alliance. We are looking forward to receiving your revised manuscript.

Sincerely,

B. MANUSCRIPT ORGANIZATION AND FORMATTING:

Reviewer #1 (Comments to the Authors (Required)):

It was long assumed that yolk production is a requirement for normal reproduction in *C. elegans*. However an earlier study from this group (Van Rompay et al 2015) showed that, surprisingly, mutants with very low levels of yolk produce a number of eggs no different to wild type. Moreover, all of the eggs hatch normally. The same study provided preliminary evidence for a possible solution to this apparent paradox: that yolk provisioning is important for survival of first stage larvae under stressful, starved conditions. This new study has at last examined this question at an appropriate level of rigor and detail. Its findings strongly support the conclusions hinted at by the earlier study: that yolk provisioning is essential for larval fitness post-hatching, i.e. for reproductive fitness. This is an important matter in terms of understanding the biology of *C. elegans* life history, not only reproduction but also aging, to which yolk production appears to be coupled. This new study brings needed clarity to the topic. *C. elegans* seems to have evolved to be able to produce their eggs even under really bad nutritional conditions, even though the quality of the progeny is lower, at least at first.

There are a few weakness in the study

1) Sections 3 and 4 are weaker than the others and really break the flow of the paper (see notes below). Fig 4 seems to me so weak that it would be best to greatly reduce reference to it, and put it in the supplement. In the proteomic analysis section, the motivation of the work is weak and the results uninteresting. I realize that it is traditional to publish profile data which doesn't really say anything, but the paper would be improved by moving it entirely to the supplement. Then the reader could quickly get to the last 2 sections of the results, which contain the real meat of the paper.

2) The work is well done, and the manuscript accurate and mostly well written. However, there are a few weakness in terms of presentation and argumentation, in terms of clarity and logical deduction. See my comments below.

5. "Yolk availability is often considered a good predictor of an oviparous species' fecundity." It is unclear whether this refers to predicting differences between species, or individuals within a species. Presumably the latter, given the nature of this study.

6. "This notion is, however, challenged..." This doesn't follow, since different organisms may differ with respect to the relationship between yolk availability and offspring number.

8-16: The deductions presented do not take into account the question of how likely it is that the behavior of mutants with reduced yolk levels reflects that of wild type *C. elegans*. Is it not possible that malnourished *C. elegans* have less yolk and lay fewer eggs, in other words that the functionally these two characteristics are normally coupled? Possibly the behavior of the mutants fails to reflect this - and shows only that it is possible by artificial means to uncouple yolk and egg production; ultimately, the purpose of studying mutants is to understand wild-type function.

18-26: The preamble to the explanation of what yolk seems too detailed; the authors risk teaching their grandmother to suck eggs.

25: "thereto" - archaic... better "for this purpose" ?

40: "necessity" - seems imprecise... yolk is not necessary for worms to lay eggs, but it is necessary for optimal fitness. Therefore there is no challenge to its necessity.

43: "four yolk proteins" - imprecise? Surely there are six yolk proteins: VIT-1 - -6, or perhaps 7 since VIT-6 is chopped in two. On a gel there are four bands.

44: pseudocoelom

56: Fig 6G in the cited study shows a slight but significant reduction in brood size. "only very modestly reduces fecundity" ?

57-65: Surely it is misleading to suggest that lack of yolk does not impair "reproductive success" or "fitness" given that work from

this group previously showed that early larvae from yolk deprived eggs are hypersensitive to starvation (Van Rompay et al 2015). Reproduction is not just a matter of egg production, but anything that contributes to production of optimally fit offspring. This appears to be a bit of a paper tiger argument.

69: phenotypic?

73-75: This seems a little off. The point is surely that it is possible, with low levels of yolk, for embryonic development to proceed sufficiently to produce larvae to hatch from eggs (though of reduced fitness). Does producing the same number of progeny of poor quality really constitute "maintaining fecundity"? Another concern is that transporters such as RME-2, in the absence of vitellogenins and therefore with extra transporting capacity, might resort to other forms of provisioning with lower affinity ligands.

Results

Fig. 1. Was the developmental delay statistically significant? Is there some reason why a statistical test was not performed here?

120: "offspring body size" - of what stage? It needs to be clear whether or not the results here are consistent with those of the Perez et al study. It slightly unclear what the significance is of the lack of effect on adult size.

Is the protein content of yolkless embryos reduced?

146: "reproductive capacity" - measured in terms of number of eggs laid; but if progeny fitness is reduced, is this not a reduction in reproductive capacity, in the meaningful sense?

149: "37% of proteins" - protein number or protein mass?

150: "greatly impact" - markedly reduce?

151: The motivation for the proteomic analysis is rather unclear. Surely the question is: is vitellogenin really superfluous for fitness? It's a yes no question. But the proteomic test seems to be asking: how does the embryo compensate for a lack of vitellogenin to ensure embryonic development?

Some things need to be said about what stage the embryos are at and how tightly synchronized development is. The less synchrony the weaker the analysis, particularly if the degree of synchrony varies between genotypes.

152: "unveil" - reveal?

167: in and of itself

169: increased or decreased (Also Table 1)

Fig 3B When one looks at the genes detected as over-represented in various categories, sometimes multiple overlaps in fact are the result of detection of the same genes. Was that done here?

207-209: The motivation of such research is unclear to me. How is it interesting to look at this? What are the compelling questions involved?

223: "no overarching conceptual logic" - no process was revealed in which the genes shared a role, that would provide new mechanistic insights ?

237: abundance

238: "contradictory" - surely not? The cells are all shouting: where's the yolk?! and upregulating their endocytic machinery in response. This could be a mechanism of compensation. What would be interesting to know is whether this is functionally important... e.g. activated in starved, wild type worms attempting to produce viable progeny.

Fig 4 Specifying the genotypes within the photos would make like easier for the reader.

232: "ceh-60 mediates endocytosis in embryos" This is a pretty weak section, better to either move to the supplement (the figure at least) or remove, or strengthen. For one there is no measurement of endocytosis, only data on distribution of an RME-2::GFP reporter expressed at levels higher than in wild type (hence the title is misleading, an over-interpretation). It is not clear whether the abnormalities in RME-2::GFP distribution are mediated by reductions in yolk levels. Could one not test this by mutation/RNAi of vitellogenin genes? But if it is, what is this telling us? That this argues against the idea of RME-2 hyperactivity in response to yolk deficiency?

297: "The state of the art scientific view subscribes" - odd phrasing - Evidence to date implies that ?

299-306: Problems again with presentation. This needs to be explained in the introduction, rather than sprung on the reader as a surprise at this point. "one could be tempted to conclude": not with an adequate introduction.

307-321: This is an important follow up to the earlier Van Rompay study.

327: multiple mutants ?

330: the other two genes ?

427: "refute", and surrounding discussion - this all seems a little overstated. *C. elegans* can manage to produce its eggs with little yolk but they are of poor quality.

430: "discovered" - overstated... earlier evidence in Van Rompay et al suggests this.

494-496: Can one say this? I could be that most of the yolk has been consumed during embryogenesis, but there is still a bit left in the gut at hatching.

506: "not always optimal" - understatement ? (!)

520-521: This is the important conclusion here. Quantity first, then, if you can manage it, quality. That worms make such efforts to make huge amounts of yolk if they can shows how important quality also is.

Reviewer #2 (Comments to the Authors (Required)):

This manuscript makes an important contribution to our understanding of the impacts of yolk provisioning on reproductive fitness in *C. elegans*. Unlike many other organisms, worms do not respond to a limited supply of yolk by limiting the number of progeny produced. However, Geens, et al. demonstrate that decreased yolk content does impact embryogenesis by extending the time needed for early embryonic cell divisions. This slower development extends through most of embryogenesis and correlates with a smaller size of L1 larvae. Once hatched larvae can feed themselves, the body size differences equalize. The authors show that the *ceh-60* mutations that decrease yolk content also impact the localization and abundance of the yolk transporter RME-2. Most interestingly, although the number of progeny does not decrease in yolk-limited worms, these progeny do appear to decrease in quality and fitness. Low embryonic yolk, and specifically YP170, correlates with L1 larvae who later have a decreased survival time when arrested by starvation. These embryonic and larval phenotypes associated with lower yolk provisioning confer an overall competitive disadvantage, confirming the importance of yolk for the overall fitness of *C. elegans*, if not for fecundity.

A few questions and suggestions:

1. The measures of early embryonic cell cycle timing (fig 1) were only done on 3 embryos. This small sample size is a little concerning but understandable due to the difficult and time-consuming nature of the experiment. However, the conclusions would be strengthened by a statistical analysis, which should be possible with the existing data.

2. I am a little concerned about the methods of collecting L1 larvae for the measurement of their size. The authors state that embryos were collected through standard bleaching and then allowed to develop for 18 hours to ensure that all embryos hatched. Because the embryos isolated by this method are mixed-stage, we would expect that they would hatch at different times within this 18 hour window. Because food was necessarily included, they would then continue to develop (and grow) through the L1 stage. Is it not possible that the different populations of gravid adult worms that were bleached might have contained different proportions of early- vs. late-stage embryos? Different genotypes are certainly known to cause this effect. Because worms continue to grow during the L1 stage of the life cycle, differences in size of measured L1s could be due to how long that L1 had been growing since hatching. A genotype or condition might appear to have smaller L1s simply because the gravid adults contained a higher proportion of earlier embryos (or embryos that developed more slowly and thus hatched later), rather than because the genotype or condition influenced the size at hatching.

3. In figure 4, I am again concerned about the very small sample size (e.g. only 3 wt worms). A larger sample size would really strengthen the conclusions. Based on the data in this figure, the authors conclude that RME-2 degradation in oocytes is impaired in *ceh-60(lst491)* mutants. There does appear to be a post-transcriptional effect since protein levels are higher but mRNA is relatively constant. However, Figure 4j shows that the higher levels of RME-2 in *ceh-60(lst491)* does decrease dramatically as the embryos develop. If anything, the slope of the curve is steeper than in wt embryos. Does that not suggest that degradation is still happening? Could the post-transcriptional effect may be something other than protein turnover?

4. For the RNAi experiments, HT115 bacteria were used. However, caution is warranted because worms fed this strain of *E. coli* have higher total fat levels than worms fed OP50. It appears that control worms were also fed L4440 in HT115, but even so, the higher baseline fat has been shown to influence germline lipid levels and therefore could affect the phenotypes described here.

Reviewer #3 (Comments to the Authors (Required)):

Geens et al follow up on their previous work analyzing *C. elegans* mutants *ceh-60* and *vrp-1* that provision very little/no vitellogenin to progeny. They note that while vitellogenin is thought to support fecundity/embryogenesis in other oviparous species, it appears dispensable in *C. elegans*, since these mutants do not display reduced brood size or embryonic inviability. Though initially surprising, this is now relatively well established by their published work and others. Here they characterize effects of these mutants on the rate of embryogenesis, body size, embryonic protein levels, RME-2 (vitellogenin receptor for endocytosis) localization and levels, embryonic fat content, L1 starvation survival, and competitive fitness in rich conditions. The work clearly shows that these mutants have compromised fitness, and the authors conclude that *C. elegans* produce offspring of lower quality when vitellogenesis is limited rather than producing fewer progeny. The phenotypic characterization of the *ceh-60* and *vrp-1* mutants is valuable, but the work is largely descriptive and lacks mechanistic insight. I also have concerns about the strength of some of the conclusions.

Major concerns

- 1) The two *ceh-60* alleles are wildly different in some phenotypic assays, (eg, L1 starvation survival, cuticle permeability, embryonic protein expression, RME-2 localization and levels). The mechanistic basis for these differences is not addressed, and it is not clear that they stem from effects on vitellogenesis in each case. However, conclusions from phenotypic analysis of *ceh-60* and *vrp-1* mutants generally assumes that the observed phenotypes originate from defective vitellogenesis, though other processes could also be affected maternally or zygotically to contribute. For starvation survival, the authors examined perturbation of various vit genes (specific yolk proteins), which strengthens the conclusions and shows that something more is clearly aberrant in *ceh-60*(*lst466*) (possibly related to its cuticle defect, but this connection is not made or addressed). A *pit-1* mutant, which has increased vitellogenesis, would have been a valuable complement as well. Unfortunately, each of the other assays presented examined only *ceh-60* and *vrp-1* mutants, and it is difficult to discern which phenotypes stem from defective vitellogenesis as opposed to other potential effects of these mutants.
- 2) The authors conclude that "*ceh-60* mediates endocytosis in embryos", using this phrase as a sub-header. However, they do not explicitly show an effect on endocytosis. The closest they come is to show decreased lipid levels in embryos in Fig S2. The conclusion is based on increased embryonic expression of proteins implicated in regulation of endocytosis, chiefly *rab-35* which is known to promote vitellogenin receptor RME-2 recycling, and the fact that RME-2 localization is disrupted in oocytes and embryos. However, loss of *rab-35* causes RME-2 to be retained in the cytoplasm and not be accumulated at the oocyte membrane (Sato 2008), and this is the phenotype they see with their *ceh-60* mutants which have increased RAB-35 levels (at least in embryos). The authors seem to suggest that endocytosis is increased, but it is unclear how these defects would increase rather than decrease it. These observations are not connected mechanistically.
- 3) The authors conclude that rather than sacrificing fecundity when vitellogenesis is reduced, *C. elegans* sacrifices progeny quality. However, the *ceh-60* and *vrp-1* mutants have huge effects on vitellogenesis and they may be pleiotropic (*ceh-60*(*lst466*) certainly is), calling physiological relevance into question. Multiple labs have shown that worms cultured in dietary restriction have reduced lipid content, vitellogenesis and progeny production, and progeny quality may also be increased. Given that DR is more ecologically relevant than the mutants studied here, the authors should reconcile these observations in their speculation.

Minor Concerns

- 1) 3 and 5 embryos were analyzed for each genotype in Fig 1A and B, but no indication is given regarding variation among embryos, how the raw data were processed, or whether means are plotted. The authors should provide these details and add statistical analysis to determine if the results are significant. Notably, the most affected *ceh-60* allele differs between the two assays, though a positive correlation is the simplest expectation, and so there is concern that these differences are simply due to small sample size.
- 2) It is unclear why two different statistical approaches were used to analyze the data in Fig 2.
- 3) Analysis of the proteomics data is unclear. With 37% less protein in the mutant embryos given a lack of VIT protein (the author's given number), the relative abundance of most other proteins should be increased compared to wild type. The fact that protein levels are increased and decreased (and with major differences between the two *ceh-60* alleles) suggests that absolute abundance was determined. However, details on normalization are not given at all. It is also unclear why or how a two-way ANOVA was used (what are the two factors, and which p values are given?) rather than, for example, t-tests.
- 4) Only 3 and 7 animals were imaged for Fig 4. As for Fig 1, inconsistent results between alleles could be simply due to small sample size.
- 5) It would be helpful to state in the text or legend that data in Fig 4 H, I are from qPCR.
- 6) Fig 5B does not include a control to show that *vit-6* RNAi worked. The authors give a pair of "interesting" interpretations for

the results, but it could just be that RNAi did not work well. This should be explicitly stated or else a protein gel should be used to demonstrate knock down as was done for the other vit perturbations presented.

7) "A *C. elegans* zygote can develop into a juvenile consisting of 671 cells in less than 12 hours (Sulston et al., 1983)." This is an inaccurate statement - there are not that many cells at hatching, and it is actually 14 hr at 25C and 18 hr at 20C (the culture temperature used).

8) "but the body size is only affected in early post-embryonic life due to hardship during embryonic development." There is no growth during embryogenesis, and size at hatching is more likely affected by the size of the oocyte than embryonic development.

Reviewer #1 (Comments to the Authors (Required)):

It was long assumed that yolk production is a requirement for normal reproduction in *C. elegans*. However an earlier study from this group (Van Rompay et al 2015) showed that, surprisingly, mutants with very low levels of yolk produce a number of eggs no different to wild type. Moreover, all of the eggs hatch normally. The same study provided preliminary evidence for a possible solution to this apparent paradox: that yolk provisioning is important for survival of first stage larvae under stressful, starved conditions. This new study has at last examined this question at an appropriate level of rigor and detail. Its findings strongly support the conclusions hinted at by the earlier study: that yolk provisioning is essential for larval fitness post-hatching, i.e. for reproductive fitness. This is an important matter in terms of understanding the biology of *C. elegans* life history, not only reproduction but also aging, to which yolk production appears to be coupled. This new study brings needed clarity to the topic. *C. elegans* seems to have evolved to be able to produce their eggs even under really bad nutritional conditions, even though the quality of the progeny is lower, at least at first.

There are a few weakness in the study

1) Sections 3 and 4 are weaker than the others and really break the flow of the paper (see notes below). Fig 4 seems to me so weak that it would be best to greatly reduce reference to it, and put it in the supplement. In the proteomic analysis section, the motivation of the work is weak and the results uninteresting. I realize that it is traditional to publish profile data which doesn't really say anything, but the paper would be improved by moving it entirely to the supplement. Then the reader could quickly get to the last 2 sections of the results, which contain the real meat of the paper.

We reorganized the text and moved most of these results, together with their description, to the revised supplemental information (Text S1 & Figure S5 (original Figure 3)). The original Figure 4 (now Figure 3) was reworked in response to a request of reviewers #2 & 3, and fits the flow of the revised body text. Overall, the manuscript now highlights the main logic of the story better, whilst still keeping all supporting experiments in the revised supplement.

2) The work is well done, and the manuscript accurate and mostly well written. However, there are a few weakness in terms of presentation and argumentation, in terms of clarity and logical deduction. See my comments below.

5. "Yolk availability is often considered a good predictor of an oviparous species' fecundity." It is unclear whether this refers to predicting differences between species, or individuals within a species. Presumably the latter, given the nature of this study.

We apologize for this statement not being clear. We rephrased the abstract (lines 5-13) to remove confusion.

6. "This notion is, however, challenged..." This doesn't follow, since different organisms may differ with respect to the relationship between yolk availability and offspring number.

*By rephrasing the beginning of the abstract to meet the previous comment, the statement of contradiction between *C. elegans* and other species is removed.*

8-16: The deductions presented do not take into account the question of how likely it is that the behavior of mutants with reduced yolk levels reflects that of wild type *C. elegans*. Is it not possible that malnourished *C. elegans* have less yolk and lay fewer eggs, in other words that the functionally these two characteristics are normally coupled? Possibly the behavior of the mutants fails to reflect this - and shows only that it is possible by artificial means to uncouple yolk and egg production; ultimately, the purpose of studying mutants is to understand wild-type function.

*The reviewer wonders whether observing the ability to uncouple yolk and embryo production in the lab, also means it really occurs in the wild. Our own competition data suggest that 'uncoupled' animals, when they appear in the wild, are unlikely to have the odds in their favor. However, interesting missense mutations in *ceh-60* are present in wild genomes (as evident from the CeNDR resource); those can hardly be called artifacts and they would certainly be worthy of exploration, in relation to natural habitat, in follow-up research. Our revised discussion now explores this idea more explicitly so, also in relation to observations made for dietary restriction (see also question by reviewer #3).*

18-26: The preamble to the explanation of what yolk seems too detailed; the authors risk teaching their grandmother to suck eggs.

We removed this part of the introduction of the revised text.

25: "thereto" - archaic... better "for this purpose" ?

Due to changes in the introduction (cf. request above), this sentence has been omitted from the revised manuscript.

40: "necessity" - seems imprecise... yolk is not necessary for worms to lay eggs, but it is necessary for optimal fitness. Therefore there is no challenge to its necessity.

We agree with the reviewer's remark and have rephrased this sentence accordingly in lines 27-29 of the revised manuscript.

43: "four yolk proteins" - imprecise? Surely there are six yolk proteins: VIT-1 - -6, or perhaps 7 since VIT-6 is chopped in two. On a gel there are four bands.

We apologize for this not being clear directly from our manuscript; it is explained in the Sharrock reference cited next to the statement. We have now added a supplementary figure (Figure S1) to our revised manuscript to explain this better, so that readers may find this information directly in our manuscript as well.

*Briefly: because of presumed gene duplications and very high sequence identity of VIT precursor proteins, the six *C. elegans vit* genes lead to only four yolk proteins: *vit-1&2* give rise to YP170B, *vit-3&4&5* to YP170A, and *vit-6* indeed to two yolk proteins, YP88 & YP115. Because both YP170 proteins (A and B) migrate to the same position, only three bands are discriminated on a standard protein gel.*

44: pseudocoelom

We changed spelling to the reviewer's preference in line 32 of the revised manuscript.

56: Fig 6G in the cited study shows a slight but significant reduction in brood size. "only very modestly reduces fecundity" ?

Indeed, this has been rephrased in line 44 of the revised manuscript.

57-65: Surely it is misleading to suggest that lack of yolk does not impair "reproductive success" or "fitness" given that work from this group previously showed that early larvae from yolk deprived eggs are hypersensitive to starvation (Van Rompay et al 2015). Reproduction is not just a matter of egg production, but anything that contributes to production of optimally fit offspring. This appears to be a bit of a paper tiger argument.

We agree that the wording in this section was not ideal and have consequently removed this section from the revised manuscript. We have, subsequently, rephrased lines 41-49 to better reflect that the motivation of this work is based on the unexpected observation of maintained egg production.

69: phenotypic?

This has been adapted in line 53 of the revised manuscript.

73-75: This seems a little off. The point is surely that it is possible, with low levels of yolk, for embryonic development to proceed sufficiently to produce larvae to hatch from eggs (though of reduced fitness). Does producing the same number of progeny of poor quality really constitute "maintaining fecundity"?

We are not sure what the reviewer struggles with in their question, because we agree that is the point, indeed. Since there is a normal amount of offspring that are viable and reproduce well, fecundity is maintained, while indeed of poorer quality.

Another concern is that transporters such as RME-2, in the absence of vitellogenins and therefore with extra transporting capacity, might resort to other forms of provisioning with lower affinity ligands.

Indeed. We had also hoped to find evidence of backup provisioning, with putative lower-affinity RME-2 ligands as one of the hypothetical entry routes, as the reviewer suggests. In fact, the concept of backup provisioning fueled our working hypothesis for the proteomics experiment (see also this reviewer's consideration with line 151), but we did not find any evidence supporting this concept.

Results

Fig. 1. Was the developmental delay statistically significant? Is there some reason why a statistical test was not performed here?

Statistical analysis is included in the corresponding supplemental Figure S2 to which we refer in the caption of Figure 1 of the main text.

120: "offspring body size" - of what stage? It needs to be clear whether or not the results here are consistent with those of the Perez et al study. It slightly unclear what the significance is of the lack of effect on adult size.

We now specify this in line 97-99 of the revised manuscript. Our observations are in line with those of Perez et al.: we both evaluated body size at L1 stage, observing smaller animals in (albeit different) cases of reduced yolk. We additionally evaluated our mutants at day 1 of adulthood (no corresponding observations in the Lehner lab study). Our revised manuscript explains the relevance of this second evaluation in lines 99-102.

Is the protein content of yolkless embryos reduced?

We cannot make full-proof claims because all current knowledge is based on measurements of protein content in young and gravid adults; with oocyte/embryo content contributing to the latter, but only as a smaller part of the total protein content. We tried several times to precisely count a few thousand eggs (using a COPAS Flow Pilot or manual approaches), but we could not confidently reach enough precisely-counted material for protein analysis. It is therefore not yet possible to state whether the non-YP proteome 'in its entirety' could be diluted or concentrated (note that if nonetheless so, this would not affect proteomic data interpretation). For (gravid) adults, we never observed major changes in the non-YP protein bands in mutants vs controls. The current view is therefore that total protein content of embryos is likely decreased due to loss of yolk proteins, but the remainder of the content probably isn't affected substantially.

146: "reproductive capacity" - measured in terms of number of eggs laid; but if progeny fitness is reduced, is this not a reduction in reproductive capacity, in the meaningful sense?

We agree with the reviewer that the use of the term 'reproductive capacity' is wrong. We corrected this in the manuscript to 'brood size' (line 109).

149: "37% of proteins" - protein number or protein mass?

In this sentence, 'proteins' refer to protein mass. We apologize that this was not clear in the initial submission. Due to rewriting of the manuscript to meet previous comments, this sentence has been removed.

150: "greatly impact" - markedly reduce?

Due to rewriting of the manuscript to meet previous comments, this sentence has disappeared.

151: The motivation for the proteomic analysis is rather unclear. Surely the question is: is vitellogenin really superfluous for fitness? It's a yes no question. But the proteomic test seems to be asking: how does the embryo compensate for a lack of vitellogenin to ensure embryonic development?

Both questions are being asked in our work, with the proteomic and follow-up experiments indeed also wondering whether any compensation might exist (see also this reviewer's question with original lines 73-75). The manuscript has now been reorganized, giving more attention to the first question in the main body text.

Some things need to be said about what stage the embryos are at and how tightly synchronized development is. The less synchrony the weaker the analysis, particularly if the degree of synchrony varies between genotypes.

Embryos were always collected from synchronized day 1 adults (68 - 72 hours after arrested L1 juveniles were placed on food). We never observed differences in mixed-embryo stage distributions when collected as such. This information is now explicitly mentioned in the main text (lines 448-450) and included in Figure S3 of the revised manuscript.

152: "unveil" - reveal?

Due to rewriting of the manuscript, this sentence has been removed.

167: in and of itself

This has been adjusted in line 14 of Text S1 of the revised manuscript.

169: increased or decreased (Also Table 1)

This has been adjusted in line 16 of Text S1 and in the caption of Table S1.

Fig 3B When one looks at the genes detected as over-represented in various categories, sometimes multiple overlaps in fact are the result of detection of the same genes. Was that done here?

We apologize that this explanation was overlooked in the initial submission. This is indeed how such analyses are done and all necessary detail has now been provided in caption of Figure S5 of the revised manuscript. As a more recent version of Panther has been released since the time of the initial submission, we reanalyzed our GO analysis and included all significant GO terms in Fig S5B.

207-209: The motivation of such research is unclear to me. How is it interesting to look at this? What are the compelling questions involved?

These differential proteins are interesting candidates for follow-up research because of their known function in metabolism and embryonic morphogenesis, but this is not the focus of our manuscript. Due to textual reorganization, these results are now part of the supplemental information (Text S1).

223: "no overarching conceptual logic" - no process was revealed in which the genes shared a role, that would provide new mechanistic insights?

This has been rephrased in lines 50-51 of the supplementary Text S1.

237: abundance

This sentence has been removed during the rewriting of the manuscript.

238: "contradictory" - surely not? The cells are all shouting: where's the yolk?! and upregulating their endocytic machinery in response. This could be a mechanism of compensation What would be interesting to know is whether this is functionally important... e.g. activated in starved, wild type worms attempting to produce viable progeny.

While we agree, severe textual reorganization has also removed this section from the revised manuscript.

Fig 4 Specifying the genotypes within the photos would make like easier for the reader.

At the reviewer's request, this has been adjusted in Figure 3 of the revised manuscript.

232: "ceh-60 mediates endocytosis in embryos" This is a pretty weak section, better to either move to the supplement (the figure at least) or remove, or strengthen. For one there is no measurement of endocytosis, only data on distribution of an RME-2::GFP

reporter expressed at levels higher than in wild type (hence the title is misleading, an over-interpretation). It is not clear whether the abnormalities in RME-2::GFP distribution are mediated by reductions in yolk levels. Could one not test this by mutation/RNAi of vitellogenin genes? But if it is, what is this telling us? That this argues against the idea of RME-2 hyperactivity in response to yolk deficiency?

This part of the manuscript has been reworked, integrating suggestions from other reviewers as well. Its title is now "ceh-60 affects proteins required for uptake and intracellular transport of yolk" and it combines the proteomics (in supplement) and updated RME-2 reporter data (in main text) of ceh-60 mutants in a more succinct manner. We kept this section focused on the ceh-60 mutants, to then move on to the use of vit mutants/RNAi in the next sections.

297: "The state of the art scientific view subscribes" - odd phrasing - Evidence to date implies that ?

This has been rephrased as suggested (line 139 of revised manuscript).

299-306: Problems again with presentation. This needs to be explained in the introduction, rather than sprung on the reader as a surprise at this point. "one could be tempted to conclude": not with an adequate introduction.

Agreed, this sentence has been removed from the revised manuscript.

307-321: This is an important follow up to the earlier Van Rompay study.

We thank the reviewer for their appreciation.

327: multiple mutants?

With 'multiple mutants', we refer to strains carrying multiple mutant genes (double and triple mutants). Because this was perceived as confusing, we removed it from the revised text. The remainder of the sentence suffices as explanation: "... we created mutants in which progressively more vit genes are deleted..." (lines 170-171 of the revised manuscript).

330: the other two genes ?

Not only other vit genes. Potentially also the other non-vit genes located at that locus. This has been clarified in the revised manuscript (line 172-174).

427: "refute", and surrounding discussion - this all seems a little overstated. C. elegans can manage to produce its eggs with little yolk but they are of poor quality.

We agree with the reviewer that this statement is too strong. This has been nuanced in lines 244-246 of the revised manuscript.

430: "discovered" - overstated... earlier evidence in Van Rompay et al suggests this.

Agreed. This has been nuanced in line 248 of the revised manuscript.

494-496: Can one say this? I could be that most of the yolk has been consumed during embryogenesis, but there is still a bit left in the gut at hatching.

This is based on Figure 7 of Bossinger & Schierenberg (1996) which do not provide a directly quantifiable readout, but clearly show that substantial amounts of yolk remain, and far from everything has been used. We rephrased the text (lines 300-

302) in our manuscript because we agree that terminology like 'most' etc. is not appropriate yet.

506: "not always optimal" - understatement ? (!)

Agreed. To say that an animal's habitat is not always optimal in an understatement. We rephrased the text accordingly (line 309-310).

520-521: This is the important conclusion here. Quantity first, then, if you can manage it, quality. That worms make such efforts to make huge amounts of yolk if they can shows how important quality also is.

We thank the reviewer for their appreciation.

Reviewer #2 (Comments to the Authors (Required)):

This manuscript makes an important contribution to our understanding of the impacts of yolk provisioning on reproductive fitness in *C. elegans*. Unlike many other organisms, worms do not respond to a limited supply of yolk by limiting the number of progeny produced. However, Geens, et al. demonstrate that decreased yolk content does impact embryogenesis by extending the time needed for early embryonic cell divisions. This slower development extends through most of embryogenesis and correlates with a smaller size of L1 larvae. Once hatched larvae can feed themselves, the body size differences equalize. The authors show that the *ceh-60* mutations that decrease yolk content also impact the localization and abundance of the yolk transporter RME-2. Most interestingly, although the number of progeny does not decrease in yolk-limited worms, these progeny do appear to decrease in quality and fitness. Low embryonic yolk, and specifically YP170, correlates with L1 larvae who later have a decreased survival time when arrested by starvation. These embryonic and larval phenotypes associated with lower yolk provisioning confer an overall competitive disadvantage, confirming the importance of yolk for the overall fitness of *C. elegans*, if not for fecundity.

A few questions and suggestions:

1. The measures of early embryonic cell cycle timing (fig 1) were only done on 3 embryos. This small sample size is a little concerning but understandable due to the difficult and time-consuming nature of the experiment. However, the conclusions would be strengthened by a statistical analysis, which should be possible with the existing data.

Results of statistical analysis have been added to Figure 1B, and in more detail also in Figure S2 for Figure 1A of the revised manuscript.

2. I am a little concerned about the methods of collecting L1 larvae for the measurement of their size. The authors state that embryos were collected through standard bleaching and then allowed to develop for 18 hours to ensure that all embryos hatched. Because the embryos isolated by this method are mixed-stage, we would expect that they would hatch at different times within this 18 hour window. Because food was necessarily included, they would then continue to develop (and grow) through the L1 stage. Is it not possible that the different populations of gravid adult worms that were bleached might have contained different proportions of early- vs. late-stage embryos? Different genotypes are certainly

known to cause this effect. Because worms continue to grow during the L1 stage of the life cycle, differences in size of measured L1s could be due to how long that L1 had been growing since hatching. A genotype or condition might appear to have smaller L1s simply because the gravid adults contained a higher proportion of earlier embryos (or embryos that developed more slowly and thus hatched later), rather than because the genotype or condition influenced the size at hatching.

This was indeed not explained in our original manuscript, for which we apologize. Embryos were always collected from synchronized day 1 adults (68 - 72 hours after arrested L1 juveniles were placed on food). We never observed differences in mixed-embryo stage distributions when collected as such. This is now explicitly mentioned in the main text (lines 94-96) and included in Figure S3 of the revised manuscript.

3. In figure 4, I am again concerned about the very small sample size (e.g. only 3 wt worms). A larger sample size would really strengthen the conclusions. Based on the data in this figure, the authors conclude that RME-2 degradation in oocytes is impaired in *ceh-60(lst491)* mutants. There does appear to be a post-transcriptional effect since protein levels are higher but mRNA is relatively constant. However, Figure 4j shows that the higher levels of RME-2 in *ceh-60(lst491)* does decrease dramatically as the embryos develop. If anything, the slope of the curve is steeper than in wt embryos. Does that not suggest that degradation is still happening? Could the post-transcriptional effect may be something other than protein turnover?

On suggestion of the reviewer, sample size has been increased and the revised Figure 3 now provides more robust support for its conclusions.

*The mentioned comparison of slopes is not possible because there is no signal left in the control conditions to base such a calculation on. The reporter data show that the RME-2 reporter signal is not cleared as expected in *lst491* embryos, which the revised manuscript states must be due to post-transcriptional effects, without speculating on particularities (lines 135-137).*

4. For the RNAi experiments, HT115 bacteria were used. However, caution is warranted because worms fed this strain of *E. coli* have higher total fat levels than worms fed OP50. It appears that control worms were also fed L4440 in HT115, but even so, the higher baseline fat has been shown to influence germline lipid levels and therefore could affect the phenotypes described here.

*While the reviewer's concern is driven by an assumption of higher fat in one condition, this depends on the life stage: higher fat levels have been observed in HT115-fed L4 larvae (Stuhr & Curran, 2020) but also in OP50-fed adults (Neve et al., 2020). Even so, it is certainly fair to wonder whether the food source may have affected our observations. Additional experiments of the revised manuscript now support that conclusions are independent of HT115 vs OP50 use: in absence of YP115/YP88 (by *vit-6* knockout) or YP170A (by *vit-1;vit-2* knockout), animals do not exhibit an aberration in competitive fitness compared to wild type worms, independent of HT115 or OP50 bacteria used during the assay (lines 232-235 and figures 5B, S8 of the revised manuscript).*

Reviewer #3 (Comments to the Authors (Required)):

Geens et al follow up on their previous work analyzing *C. elegans* mutants *ceh-60* and *vrp-1* that provision very little/no vitellogenin to progeny. They note that while vitellogenin is thought to support fecundity/embryogenesis in other oviparous species, it appears dispensable in *C. elegans*, since these mutants do not display reduced brood size or embryonic inviability. Though initially surprising, this is now relatively well established by their published work and others. Here they characterize effects of these mutants on the rate of embryogenesis, body size, embryonic protein levels, RME-2 (vitellogenin receptor for endocytosis) localization and levels, embryonic fat content, L1 starvation survival, and competitive fitness in rich conditions. The work clearly shows that these mutants have compromised fitness, and the authors conclude that *C. elegans* produce offspring of lower quality when vitellogenesis is limited rather than producing fewer progeny. The phenotypic characterization of the *ceh-60* and *vrp-1* mutants is valuable, but the work is largely descriptive and lacks mechanistic insight. I also have concerns about the strength of some of the conclusions.

Major concerns

1) The two *ceh-60* alleles are wildly different in some phenotypic assays, (eg, L1 starvation survival, cuticle permeability, embryonic protein expression, RME-2 localization and levels). The mechanistic basis for these differences is not addressed, and it is not clear that they stem from effects on vitellogenesis in each case. However, conclusions from phenotypic analysis of *ceh-60* and *vrp-1* mutants generally assumes that the observed phenotypes originate from defective vitellogenesis, though other processes could also be affected maternally or zygotically to contribute. For starvation survival, the authors examined perturbation of various vit genes (specific yolk proteins), which strengthens the conclusions and shows that something more is clearly aberrant in *ceh-60*(*lst466*) (possibly related to its cuticle defect, but this connection is not made or addressed). A *pitr-1* mutant, which has increased vitellogenesis, would have been a valuable complement as well. Unfortunately, each of the other assays presented examined only *ceh-60* and *vrp-1* mutants, and it is difficult to discern which phenotypes stem from defective vitellogenesis as opposed to other potential effects of these mutants.

*lst466, lst491 and lst539 mutants indeed differ quite a lot from each other, apart from sharing the reduced vit phenotype. We fully agree that the shared phenotypes observed here can, but do not necessarily have to, be due to the shared reduction in vitellogenesis. Bearing in mind that an all-vit mutant cannot be made (see manuscript text), we combined observations for these mutants to help us build hypotheses on possible effects of yolk deprivation, for which we then made new, precise mutants to test in functional assays (L1 starvation survival and competitive fitness). While increased vitellogenesis could indeed also be interesting in competition context, we could not rely on *pitr-1* for these assays, because the animals have several other phenotypes, including reduced brood size, which would confound the interpretation. We hope the conceptual logic and corresponding conclusions are unambiguously clear in the revised manuscript text.*

2) The authors conclude that "ceh-60 mediates endocytosis in embryos", using this phrase as a sub-header. However, they do not explicitly show an effect on endocytosis. The closest they come is to show decreased lipid levels in embryos in Fig S2. The conclusion is based on increased embryonic expression of proteins implicated in regulation of endocytosis, chiefly rab-35 which is known to promote vitellogenin receptor RME-2 recycling, and the fact that RME-2 localization is disrupted in oocytes and embryos. However, loss of rab-35 causes RME-2 to be retained in the cytoplasm and not be accumulated at the oocyte membrane (Sato 2008), and this is the phenotype they see with their ceh-60 mutants which have increased RAB-35 levels (at least in embryos). The authors seem to suggest that endocytosis is increased, but it is unclear how these defects would increase rather than decrease it. These observations are not connected mechanistically.

Agreed. This part of the manuscript has been reorganized to rather focus on the proteomics (supplement Text S1) and RME-2 reporter data of both ceh-60 mutants.

3) The authors conclude that rather than sacrificing fecundity when vitellogenesis is reduced, *C. elegans* sacrifices progeny quality. However, the ceh-60 and vrp-1 mutants have huge effects on vitellogenesis and they may be pleiotropic (ceh-60(Ist466) certainly is), calling physiological relevance into question. Multiple labs have shown that worms cultured in dietary restriction have reduced lipid content, vitellogenesis and progeny production, and progeny quality may also be increased. Given that DR is more ecologically relevant than the mutants studied here, the authors should reconcile these observations in their speculation.

Next to observations in ceh-60 and vrp-1 mutants, other yolk-reducing interventions, including vit RNAi, do not affect offspring numbers in C. elegans (e.g. Gems, Lehner, Lapierre labs, amongst others). We here show that this nonetheless comes with a cost to progeny quality, and therefore fitness.

That being said, the reviewer probably more generally wonders whether vit reduction in nature ever happens without a matching reduction of progeny production. This is something descriptive studies of the future will need to tell: it could be studied by relying only on wild isolates or wild types, tested under different nature-mimicking conditions. DR would be one of those, and supporting the notion the reviewer makes, at least one form of DR (it comes in many mechanistically and phenotypically different flavors) reduces vit levels in adults (Seah et al., 2015) but increases VIT content of embryos (Jordan et al., 2019), which is thought to be due to a simultaneously reduced embryo production under DR. However, other situations may have other outcomes, and we already know there are other factors to consider as well – for example, yolk content of embryos also differs depending on maternal age.

*While the question is interesting, it is not the scope of our manuscript. Our experiments support a functional-genetic organization of *C. elegans* that enables to uncouple yolk and egg production under standard laboratory conditions, and entails certain implications for competitive fitness. Whether specific natural or environmental interventions exploit this, is a very interesting question but will only become clear through future research beyond the Petri dish.*

Minor Concerns

1) 3 and 5 embryos were analyzed for each genotype in Fig 1A and B, but no indication is given regarding variation among embryos, how the raw data were processed, or whether means are plotted. The authors should provide these details and add statistical analysis to determine if the results are significant. Notably, the most affected *ceh-60* allele differs between the two assays, though a positive correlation is the simplest expectation, and so there is concern that these differences are simply due to small sample size.

Variation and details are now provided in Figure S2 (detailed representation of data used presented in Figure 1A) and Figure 1B of the revised manuscript. While, as recognized by another reviewer, feasible sample numbers are limited by the complex and demanding nature of the experiments, the reviewer's observation is correct. Experimental setups differ quite a bit between Fig 1A/B assays, which means there may be unknown contributors to allele-specific effects. Together, our data permit to conclude that there is only a small effect on the timing of embryonic development at best, in these mutants.

2) It is unclear why two different statistical approaches were used to analyze the data in Fig 2.

To clarify the reasoning behind the choice of the tests used during our statistical analysis, we included a 'Statistical analysis' section in the 'Materials and methods'. In the case of Figure 2, different statistical approaches were used since data presented in the right graph of panel B are normally distributed while this is not the case for the data shown in the other graphs.

3) Analysis of the proteomics data is unclear. With 37% less protein in the mutant embryos given a lack of VIT protein (the author's given number), the relative abundance of most other proteins should be increased compared to wild type. The fact that protein levels are increased and decreased (and with major differences between the two *ceh-60* alleles) suggests that absolute abundance was determined. However, details on normalization are not given at all. It is also unclear why or how a two-way ANOVA was used (what are the two factors, and which p values are given?) rather than, for example, t-tests.

*We apologize for details on normalization not being clear in the initial submission. Normalization was done in relative terms, and essentially as per CONSTAND (Maes et al., 2016; now added to lines 478-480 of the revised manuscript). Because proteins are weighed in this process, the effects pointed out as a risk by the reviewer are avoided. As part of the normalization workflow that controls for this, it is verified that there is no effect of removal of differential proteins out of the set on normalization. Regarding the different mutants: the vast majority of proteins is not differentially present in either mutant. Based on other observations by us and others (Van de Walle et al., 2019 & Downen, 2019) we speculate that the clear differences between *ceh-60* alleles (Figure S5A of the revised manuscript) may be explained by the behavior of this protein in its transcription factor complex context. However, because it is highly speculative at this moment, we chose this idea is not yet ready for wider communication.*

*A two-way ANOVA with Benjamini & Hochberg multiple comparison correction was used to analyze the proteomics data since a comparison was made between the abundance of a protein (N = 6 corresponding to six TMT labels) between genotypes. The P values given in Table S1, consequently, correspond to the performed post hoc test for each *ceh-60* mutant.*

4) Only 3 and 7 animals were imaged for Fig 4. As for Fig 1, inconsistent results between alleles could be simply due to small sample size.

We increased the sample size in response to a request of reviewers #2 & #3 and reworked the manuscript text and relevant figure (Figure 3 of the revised manuscript) accordingly.

5) It would be helpful to state in the text or legend that data in Fig 4 H, I are from qPCR.

As per reviewer's request, we now mention that the results plotted in these figures (Fig 3C & 3F in the revised manuscript) were obtained using qPCR.

6) Fig 5B does not include a control to show that vit-6 RNAi worked. The authors give a pair of "interesting" interpretations for the results, but it could just be that RNAi did not work well. This should be explicitly stated or else a protein gel should be used to demonstrate knock down as was done for the other vit perturbations presented.

Abolishment of YP115/YP88 was established via a full knockout of vit-6 and not via RNAi. This knockout was verified via PCR before the start of the experiment. In case the reviewer meant vit-5 instead of vit-6 RNAi: Figure S7 shows that vit-5 RNAi-treated worms clearly have a significantly reduced YP170 pool.

7) "A *C. elegans* zygote can develop into a juvenile consisting of 671 cells in less than 12 hours (Sulston et al., 1983)." This is an inaccurate statement - there are not that many cells at hatching, and it is actually 14 hr at 25C and 18 hr at 20C (the culture temperature used).

The Sulston observations indeed state 800 min, which is 13.3 hours (and not "less than 12"). We updated this in the manuscript, but this still differs from what the reviewer states. Since we are not aware of a literature source that contests this, we rely on the Sulston reference for cited numbers in line 273 of the revised manuscript. Despite this, all can probably agree that embryogenesis comes with a quick rise in cell number, which is the point that is being made.

8) "but the body size is only affected in early post-embryonic life due to hardship during embryonic development." There is no growth during embryogenesis, and size at hatching is more likely affected by the size of the oocyte than embryonic development.

Agreed. We removed that suggestion out of the revised manuscript.

March 21, 2023

RE: Life Science Alliance Manuscript #LSA-2022-01675R

Ms. Ellen Geens
KU Leuven
Naamsestraat 59
Leuven 3000
Belgium

Dear Dr. Geens,

Thank you for submitting your revised manuscript entitled "Yolk-deprived *Caenorhabditis elegans* secure brood size at the expense of competitive fitness". We would be happy to publish your paper in Life Science Alliance pending final revisions necessary to meet our formatting guidelines.

- please add the Twitter handle of your host institute/organization as well as your own or/and one of the authors in our system
- please add a figure callout for Figure S4 and Figure S5A to your main manuscript text
- please include the accession info for the mass spectrometry data files in your Data Availability Statement
- the information and References in the Text S1 file should be incorporated into the main manuscript and Reference list

A. FINAL FILES:

B. MANUSCRIPT ORGANIZATION AND FORMATTING:

Sincerely,

Reviewer #2 (Comments to the Authors (Required)):

This revised manuscript shows that, although *C. elegans* yolk provisioning does not increase fecundity, it nonetheless benefits progeny by decreasing time of embryogenesis and increasing L1 larval size. Importantly, this translates into an increase in the quality and stress resistance of the progeny.

The authors have adequately addressed prior concerns regarding the manuscript.

Reviewer #3 (Comments to the Authors (Required)):

The manuscript by Geens et al is much improved in revision, and the authors have addressed my concerns.

As for the number of cells generated during embryogenesis, I was thinking of the number of cells in the L1 at hatching, which is less than the number generated given programmed cell death.

March 29, 2023

RE: Life Science Alliance Manuscript #LSA-2022-01675RR

Ms. Ellen Geens
KU Leuven
Naamsestraat 59
Leuven 3000
Belgium

Dear Dr. Geens,

Thank you for submitting your Research Article entitled "Yolk-deprived *Caenorhabditis elegans* secure brood size at the expense of competitive fitness". It is a pleasure to let you know that your manuscript is now accepted for publication in Life Science Alliance. Congratulations on this interesting work.

DISTRIBUTION OF MATERIALS:

Again, congratulations on a very nice paper. I hope you found the review process to be constructive and are pleased with how the manuscript was handled editorially. We look forward to future exciting submissions from your lab.

Sincerely,
